# Comparative Genome Analysis of Polar *Mesorhizobium* sp. PAMC28654 to Gain Insight into Tolerance to Salinity and Trace Element Stress

**DOI:** 10.3390/microorganisms12010120

**Published:** 2024-01-07

**Authors:** Anamika Khanal, So-Ra Han, Jun Hyuck Lee, Tae-Jin Oh

**Affiliations:** 1Genome-Based Bio-IT Convergence Institute, Asan 31460, Republic of Korea; anamika.khanal@gmail.com (A.K.); 553sora@hanmail.net (S.-R.H.); 2Bio Big Data-Based Chungnam Smart Clean Research Leader Training Program, SunMoon University, Asan 31460, Republic of Korea; 3Department of Life Science and Biochemical Engineering, Graduate School, SunMoon University, Asan 31460, Republic of Korea; 4Research Unit of Cryogenic Novel Materials, Korea Polar Research Institute, Incheon 21990, Republic of Korea; junhyucklee@kopri.re.kr; 5Department of Pharmaceutical Engineering and Biotechnology, SunMoon University, Asan 31460, Republic of Korea

**Keywords:** cold adaptation, comparative genomics, exopolysaccharide, *Mesorhizobium* sp. PAMC28654, salinity, trace element

## Abstract

In this study, *Mesorhizobium* sp. PAMC28654 was isolated from a soil sample collected from the polar region of Uganda. Whole-genome sequencing and comparative genomics were performed to better understand the genomic features necessary for *Mesorhizobium* sp. PAMC28654 to survive and thrive in extreme conditions and stresses. Additionally, diverse sequence analysis tools were employed for genomic investigation. The results of the analysis were then validated using wet-lab experiments. Genome analysis showed trace elements’ resistant proteins (CopC, CopD, CzcD, and Acr3), exopolysaccharide (EPS)-producing proteins (ExoF and ExoQ), and nitrogen metabolic proteins (NarG, NarH, and NarI). The strain was positive for nitrate reduction. It was tolerant to 100 mM NaCl at 15 °C and 25 °C temperatures and resistant to multiple trace elements (up to 1 mM CuSO_4_·5H_2_O, 2 mM CoCl_2_·6H_2_O, 1 mM ZnSO_4_·7H_2_O, 0.05 mM Cd(NO_3_)_2_·4H_2_O, and 100 mM Na_2_HAsO_4_·7H_2_O at 15 °C and 0.25 mM CuSO_4_·5H_2_O, 2 mM CoCl_2_·6H_2_O, 0.5 mM ZnSO_4_·7H_2_O, 0.01 mM Cd(NO_3_)_2_·4H_2_O, and 100 mM Na_2_HAsO_4_·7H_2_O at 25 °C). This research contributes to our understanding of bacteria’s ability to survive abiotic stresses. The isolated strain can be a potential candidate for implementation for environmental and agricultural purposes.

## 1. Introduction

The *Mesorhizobium* genus was identified by Jarvis et al. in 1997 [1] and is a member of the family *Phyllobacteriaceae*, phylum *Pseudomonadota*, and class *Alphaproteobacteria* [2,3]. The *Mesorhizobium* species are Gram-negative, aerobic, and non-spore-forming bacteria. They have been isolated worldwide from terrestrial environments, especially from soil and root nodules. They have also been isolated from seawater, wastewater treatment systems, and groundwater [4,5,6]. The broad distribution of this genus and its ability to form a symbiotic relationship with several plant genera make it an interesting candidate from agronomic and ecological perspectives [7,8]. In addition, exopolysaccharides (EPSs) produced by rhizobium such as *Mesorhizobium* have contributed significantly to various industries, such as gelling, thickening, and stabilizing agents in foods, pharmaceuticals, and cosmetics [9]. EPS-producing bacteria might have the potential to manage multiple abiotic stresses such as temperature stress, heavy metal (HM) stress, and salinity stress through an adaptive strategy [10].

Polar regions have an isolated environment and an extremely harsh climate. It has been reported that polar regions are affected by increased human activities including climate change and global warming. Furthermore, abiotic stresses such as HMs, temperature stress, salinity, and nutrient availability in polar areas have been reported. Microorganisms living in these areas have developed adaptive strategies to survive in extreme conditions and resist various abiotic stresses [11].

Temperature is one of the biggest factors that can influence the growth and survival of bacteria. Bacteria have been reported to tolerate a wide range of temperatures from extremely high temperatures (thermophiles) to extremely low temperatures (psychrophiles). Bacteria can mediate their ability to tolerate low or cold temperatures by performing structural adjustments of enzymes, maintaining membrane fluidity, expressing cold shock proteins, and adapting translation and transcription machinery [10,12]. In addition to temperature stress, trace element stress, which includes heavy metals and metalloids such as copper, Cu; zinc, Zn; cadmium, Cd; arsenate, As; lead, Pb; mercury, Hg; chromium, Cr; and nickel, Ni, is another factor that can have a huge influence on bacterial survival. It has been reported that certain microorganisms (such as *Acinetobacter*, *Arthrobacter*, *Bacillus*, *Agrobacterium*, *Dyadobacter*, *Enterobacter*, *Exiguobacterium*, *Kluyvera*, and *Micrococcus* including *Mesorhizobium metallidurans*) collected from areas contaminated by HMs have evolved sophisticated mechanisms to deal with HMs, which can help them grow in an environment contaminated by HMs [13,14,15]. In addition to temperature and HM stresses, salt stress is another factor that has a huge impact on bacterial growth and survival. It is the biggest hurdle in achieving better crop yield and quality with high-salinity soil [16]. Therefore, it is very important to explore rhizobacteria with the ability to adapt to salt-stress environments and apply them to achieve better crop yield in an environment with salt stress. Studying the diversity of these bacteria has various advantages in tackling issues of salinity and HM pollution to secure a sustainable future. Although *Mesorhizobium* species have been studied mostly at optimum temperatures (25 °C–30 °C) [17] for agronomic, ecological, and industrial applications [18], very limited studies have been carried out on *Mesorhizobium* bacteria in extreme environmental conditions such as low temperatures, trace element stress, and salinity.

The main aim of this study was to isolate a *Mesorhizobium* strain from cold regions and study their survival possibilities in cold areas. This study also aimed to determine specific genes and some special features of the isolated strain in terms of their tolerance to trace elements and salinity. This study is very helpful for obtaining a better understanding of how bacteria can mediate their survival under cold temperatures and other stresses, such as salinity and trace element stresses, based on the production of EPS. The isolated strain might have environmental and agricultural applications in the future.

## 2. Materials and Methods

### 2.1. Isolation and Genomic DNA Extraction of Mesorhizobium sp. PAMC28654

The strain *Mesorhizobium* sp. PAMC28654 was isolated from soil samples collected in the Rwenzori Mountains of Uganda (latitude 00°23′09″ N and longitude 29°52′18″ E) by the Korea Polar Research Institute (KOPRI, Incheon, Republic of Korea). After performing serial dilution by diluting soil samples with sterile double-distilled water, samples were added to a 0.1 X R2A plate (MB cell Ltd., Seoul, Republic of Korea) and incubated at a temperature of 10 °C for 10 days. The serial dilution was performed to separate colonies far apart in the isolation plate. An isolated single colony was then aseptically picked and transferred to an R2A agar medium using the streak plate method to obtain a pure colony. The morphology of the colony was observed. Genomic DNA was then extracted from *Mesorhizobium* sp. PAMC28654 using a QIAamp DNA Mini Kit (Qiagen Inc., Valencia, CA, USA). The quantity and purity of the genomic DNA were assessed using a spectrophotometer (Biochrome, Libra S35PC, Cambridge, UK). The extracted DNA was further evaluated for quality through agarose gel electrophoresis and subsequently stored at −20 °C for future genome analysis.

### 2.2. Genome Sequencing and Assembly Process

Before initiating the genome sequencing process, the quantity and purity of the genomic DNA were determined using an Agilent 2100 Bioanalyzer (Agilent Technologies, Santa Clara, CA, USA). This step was performed to ensure the quality of the genomic DNA for obtaining a complete genome sequence. Genome sequencing was conducted using the PacBio RS II single-molecule real-time (SMRT) sequencing technology from Pacific Biosciences (Menlo Park, CA, USA). SMRTbell library inserts of 20 kb were prepared and sequenced using SMRT cells. Raw sequencing data were generated and subjected to de novo assembly utilizing the hierarchical genome assembly process (HGAP) protocol [19] and RS HGAP4 Assembly in SMRT analysis software (ver. 2.3; Pacific Biosciences, SMRT Link 4.0.0). 

### 2.3. Functional Annotation and Comparative Genomics Analysis with Mesorhizobium sp. PAMC28654

In this study, a diverse set of annotation tools was employed to comprehensively investigate the genomic characteristics of the strain PAMC28654. Annotation of the genome was performed using the NCBI Prokaryotic Genome Annotation Pipeline (PGAP). Detailed information about PGAP can be found at https://www.ncbi.nlm.nih.gov/genome/annotation_prok/ (accessed on 18 October 2021). Initially, the genome of PAMC28654 was annotated with rapid annotation using the Subsystem Technology (RAST) server [20] available at https://rast.nmpdr.org/ (accessed on 21 January 2022). Furthermore, were predicted. The RAST server enables the identification and annotation of genes along with their associated functions as well as coding DNA sequences (CDSs). Subsequently, predicted gene sequences were translated and subjected to an extensive search across multiple databases, including the National Center for Biotechnology Information (NCBI) non-redundant database, UniProtKB/Swiss-Prot, and Protein Data Bank proteins (PDB). This comprehensive annotation approach allowed us to gather a thorough understanding of the genomic features of the strain PAMC28654. By leveraging multiple databases and annotation tools, we could validate and enhance the annotations, ensuring the reliability and accuracy of identified genes and their potential functions. By employing this meticulous annotation strategy, we unveiled crucial insights into the genetic makeup of the strain PAMC28654 and gained a deeper understanding of its biological potential and function attributes. The complete genome of the *Mesorhizobium* sp. PAMC28654 was visualized using the CGView server in Proksee (https://proksee.ca/), accessed on 2 February 2022, to generate a circular map [21].

Similarly, we obtained 61 complete genomes of *Mesorhizobium* strains from the NCBI database. These genomes were downloaded and utilized for comparative analysis to investigate the unique characteristics and variations within the species. We conducted a comparative genome analysis to gain insights into these genomes’ genetic diversity, evolutionary relationships, and distinctive features. This comparative approach allowed us to explore the broader genomic landscape of this species, facilitating a comprehensive understanding of its genomic characteristics.

### 2.4. Phylogenetic Analysis

Phylogenetic trees were constructed using 16S rRNA sequences of complete genomes of uncategorized and categorized *Mesorhizobium* strains obtained from NCBI. These 16S rRNA sequences were aligned using MUSCLE, MEGA 11, and a neighbor-joining method [22,23,24,25,26,27]. A maximum composite likelihood model was used to construct the phylogenetic tree. Branch numbers represent the percentages of bootstrap values in 1000 sampling replicates. Related whole-genome sequences of *Mesorhizobium* strains available in GenBank (https://www.ncbi.nlm.nih.gov), accessed on 30 March 2023, were also downloaded for identification and comparison with our strain. Genome-based taxonomic analysis was performed using a free bioinformatics platform Type (Strain) Genome Server (TYGS (https://tygs.dsmz.de), accessed on 20 November 2023 [28]. Based on TYGS results, the similarity between strains was confirmed by comparing values of OrthoANI, which were calculated using the Orthologous Average Nucleotide Identity Tool (OAT) [29], can be found at http://ezbiocloud.net/tools/ani (accessed on 3 November 2023).

### 2.5. Bacterial Growth in Different Media for Wet-Lab Experiments

The isolated strain was checked for their growth in several media, including R2A, lysogeny agar (LBA), nutrient agar (NA), tryptic soy agar (TSA), and marine agar (MA), at three different temperatures of 15 °C, 25 °C, and 37 °C. The strain was also tested for growth in nitrate broth containing peptone (5 g), meat extract (3 g), and KNO_3_ (1 g) with pH 7.0 at three different temperatures of 15 °C, 25 °C, and 37 °C.

### 2.6. Nitrogen Metabolism, 3D Modeling of Nitrate Reductase, and Nitrate Reduction Assay (Qualitative Test)

The nitrogen metabolic pathway was constructed using the Kyoto Encyclopedia of Genes and Genomes (KEGG) database and BlastKOALA. The putative 3D structure of nitrate reductase was modeled using the online program PHYRE2 Server (http://genome3d.eu/), accessed on 12 September 2023, in an intensive mode based on the most identical template in the PDB database [30]. Nitrate reduction tests were performed using a method based on the ability of bacteria to reduce nitrate to nitrite. The presence of nitrite can be detected with specific reagents such as reagent A (sulfanilic acid 8 g/L, glacial acetic acid 286 mL/L, and demineralized water 714 mL/L) and reagent B (glacial acetic acid 286 mL/L, N,N-dimethyl-1-napthaylamine 6 mL/L, and demineralized water 714 mL/L), which could produce changes in color. For confirmation of nitrate reduction, zinc dust was also added. The strain *Mesorhizobium* sp. PAMC28654 was cultured in a nitrate broth at three different temperatures, 15 °C, 25 °C, and 37 °C, for 2 days. An abiotic control without any microorganism and a positive control with an *Escherichia coli* strain were also included. After 1 ml of bacterial culture grown at each temperature was taken into a test tube, a few drops of reagent A and reagent B were added. The change in color from a colorless solution to red was monitored. For confirmation, a pinch of zinc dust was added to the tube with reagents A and B.

### 2.7. EPS Production and Scanning Electron Microscopy (SEM) Analysis

The isolated strain was tested for EPS production at both temperatures of 15 °C and 25 °C. To test for EPS production by the bacterium, a colony of strain *Mesorhizobium* sp. PAMC28654 was inoculated into 100 mL of R2A broth, followed by incubation in a shaking incubator at 15 °C and 25 °C for 7 days. EPS was extracted as described previously [31]. The culture was then centrifuged at 13,000× *g* for 30 min at 4 °C. After that, 96% ethanol (1:3 *v*/*v*) was added. The mixture was kept under refrigeration at 4 °C for 24 h. Samples were then centrifuged again at 13,000× *g* for 30 min at 4 °C. The pellet was dried and its weight was measured. The surface morphology of the extracted EPSs was visualized using a field emission scanning electron microscope (FE-SEM) (JEOL, JSM-6700F).

### 2.8. Salinity Test

Salinity tests of bacteria were performed at two different temperatures of 15 °C and 25 °C using different concentrations (500 mM, 100 mM, 50 mM, 10 mM, 5 mM, 2.5 mM, 2 mM, 1 mM, 0.5 mM, 0.25 mM, 0.1 mM, 0.05 mM, and 0.01 mM) of NaCl. Furthermore, for the salinity experiment, (1) control (*Mesorhizobium* sp. PAMC28654 bacterial cells were added in a culture tube containing media but without NaCl) and (2) test (*Mesorhizobium* sp. PAMC28654 bacterial cells were added in a culture tube containing media with NaCl) and were incubated at two different temperatures of 15 °C and 25 °C for 7 days. Bacterial growth was monitored by measuring OD_600_ with a spectrophotometer. The initial concentration of bacterial cells used for determining salinity toxicity was ~2.42 × 10^8^ cells/mL (reference *E. coli* cells, OD_600_ of 1.0 = 8 × 10^8^ cells/mL). Experiments were performed in triplicate.

### 2.9. Toxicity of Trace Elements

Trace element toxicity was evaluated at two different temperatures of 15 °C and 25 °C using the salts of trace elements such as copper sulfate pentahydrate (CuSO_4_·5H_2_O), cobalt chloride hexahydrate (CoCl_2_·6H_2_O), zinc sulfate heptahydrate (ZnSO_4_·7H_2_O), and cadmium nitrate tetrahydrate (Cd(NO_3_)_2_·4H_2_O) at concentrations of 10 mM, 5 mM, 2 mM, 1 mM, 0.5 mM, 0.25 mM, 0.1 mM, 0.05 mM, and 0.01 mM and disodium hydrogen arsenate heptahydrate (Na_2_HAsO_4_·7H_2_O) at different concentrations of 500 mM, 300 mM, 200 mM, 100 mM, 50 mM, 10 mM, 5 mM, 2 mM, 1 mM, 0.5 mM, 0.25 mM, 0.1 mM, 0.05 mM, and 0.01 mM. These prepared solutions were filtered using sterilized 0.2 μm filters. Furthermore, for the trace elements experiment, (1) control (*Mesorhizobium* sp. PAMC28654 bacterial cells were added in a culture tube containing media but without trace elements) and (2) test (*Mesorhizobium* sp. PAMC28654 bacterial cells were added in a culture tube containing media with trace elements) tubes were incubated at two different temperatures of 15 °C and 25 °C for 7 days. Bacterial growth was monitored by measuring OD_600_ with a spectrophotometer. The initial concentration of bacterial cells used for determining trace element toxicity was ~2.42 × 10^8^ cells/mL (reference *E. coli* cells, OD_600_ of 1.0 = 8 × 10^8^ cells/mL). Experiments were performed in triplicate.

## 3. Results and Discussion

### 3.1. Complete Genome Information of Mesorhizobium sp. PAMC28654 and Comparison of Genome Information from Closely Related Species

The complete genome sequence of *Mesorhizobium* sp. PAMC28654 was successfully deposited in the NCBI database with an accession number of NZ_CP076547.1. As indicated in Table 1, the genome of strain PAMC28654 is composed of a single circular chromosome with a length of 6,701,426 base pairs (bp) and a GC content of 62.2%. Assembling the genome resulted in a single contig. Through comprehensive gene prediction analysis, a total of 6642 genes were identified from the chromosome. Of them, 6178 genes were annotated as protein-encoding genes with known functions. Additionally, the genome harbored 400 pseudogenes, 6 rRNA genes, 54 tRNA genes, and 4 other RNA genes, which were found to be distributed throughout the genome. The genome information of *Mesorhizobium* sp. PAMC28654 and the circular map are shown in Table 1 and Figure 1, respectively. Subsystem distribution based on RAST SEED analysis of *Mesorhizobium* sp. PAMC28654 predicted to be involved in the cellular process is shown in Figure 2.

In the NCBI database, a comprehensive search revealed that a total of 61 complete genomes of the *Mesorhizobium* genus had been deposited to date. Among them, *Mesorhizobium* sp. NBSH29 exhibited a relatively low GC percentage of 58.96%, whereas *Mesorhizobium* sp. 8 displayed a high GC percentage of 65.2%. On the other hand, *Mesorhizobium* sp. PAMC28654 demonstrated a GC percentage of 62.2%, a range that was neither particularly high nor low. Detailed information is given in Appendix A. These variations in GC content among different strains of the *Mesorhizobium* genus could contribute to genomic diversity within the species. They might hold significance in terms of their adaptive traits and evolutionary history.

### 3.2. Phylogenetic Analysis

A phylogenetic tree was constructed using 16S rRNA sequences of complete genomes of *Mesorhizobium* strains obtained from NCBI using MEGA 11. The results showed that the *Mesorhizobium* sp. PAMC28654 strain was most closely related to *Mesorhizobium* sp. INR15 (CP045496.1) [32] with a similarity of 99.87%, *Mesorhizobium* sp. NZP2077 (CP051293.1) (unpublished data) with a similarity of 99.80%, and *Mesorhizobium* loti R88b (CP033367.1) [33] with a similarity of 99.66% (Figure 3A). Additional attempts were made to evaluate the *Mesorhizobium* sp. PAMC28654 strain using genome-based taxonomy. The whole-genome sequence of the *Mesorhizobium* sp. PAMC28654 strain was compared to whole genomes of all *Mesorhizobium* strains from a database of type strains using the MASH algorithm. The set of type strains with the smallest MASH distance to the *Mesorhizobium* sp. PAMC28654 genome was observed/selected from TYGS [28]. Furthermore, a genome blast distance phylogeny (GBDP) approach was used to perform a pairwise comparison of strains that were closely related to the *Mesorhizobium* sp. PAMC28654 strain so that the exact inter-genome distance could be inferred based on the trimming algorithm and distance. The results are shown in Figure 3B. The comparison of average nucleotide identity (ANI) values between a total of 19 strains, including *Mesorhizobium* sp. PAMC28654, the closest type lineage determined from the TYGS database, was carried out to determine bacterial species identification between the *Mesorhizobium* sp. PAMC28654 strain and the selected reference strains. Genome sequences’ ANI were found to range from 83.92% to 99.99% (Figure 3C). However, the ANI value obtained with the complete genome of *Mesorhizobium* sp. PAMC28654 strain was much lower than the typical ANI value of 96%. Furthermore, the ANI value of the *Mesorhizobium* sp. PAMC28654 strain was significantly lower than 96% when it was compared to other strains. Thus, the *Mesorhizobium* sp. PAMC28654 strain was not close to other strains. All bacterial orthologous genes shared an average nucleotide identity across any two genomes, according to ANI analysis. It provides a strong identification and resolution between bacterial strains of the same or closely related species (i.e., species displaying ANI values over 96%) and between bacterial strains of the same or closely related species (i.e., species displaying 80–100% ANI) [34,35]. These results showed that the *Mesorhizobium* sp. PAMC28654 strain might belong to a new *Mesorhizobium* species. This strain was deposited in the NCBI database as *Mesorhizobium* sp. PAMC28654. 

### 3.3. Bacterial Growth

Growth of the isolated strain *Mesorhizobium* sp. PAMC28654 was observed in R2A broth and nutrient agar (NA) at two different temperatures of 15 °C and 25 °C. In addition, growth was also observed in specific media such as nitrate broth and nitrite broth at 15 °C and 25 °C. No growth of the strain *Mesorhizobium* sp. PAMC28654 was observed at 37 °C. Thus, no further experiment was performed at 37 °C.

### 3.4. Nitrogen Metabolism, 3D Models of Nitrate Reductase Enzyme, and Nitrate Reduction Assay

Genome analysis of *Mesorhizobium* sp. PAMC28654 showed the presence of nitrogen metabolic enzymes, transcription factors, and transporters (Figure 4). Among three different nitrate-reducing systems (Nas, Nar, and Nap) reported in prokaryotes [36,37], membrane-bounded nitrate reductases (Nar) with all three subunits (NarG, α-subunit; NarH, β-subunit; and NarI, γ-subunit) have been identified in *Mesorhizobium* PAMC28654. Nar enzymes have been reported to be responsible for the conversion of the initial step in the nitrate reduction pathway and often in an anaerobic or low-oxygen condition [36,37]. NarG is responsible for binding and reducing nitrate to nitrite, which can facilitate a reduction in nitrate by transferring electrons to NarG and NarI, anchoring NarG and NarH subunits to the bacterial membrane. In addition, nitrate/nitrite protein (NarK) is responsible for importing nitrate from the extracellular environment into bacterial cells to be utilized by the nitrate reductase complex. Putative 3D structure modeling of the three subunit Nar enzymes (NarG, α-subunit; NarH, β-subunit; and NarI, γ- subunit) are shown in Appendix A. Putative 3D structure modeling of (1) NarG was generated based on sequence residues from 2 to 942 with 92% coverage, 100% confidence, and 67% identity against C1Y5iA (1.90 Å). Putative 3D structure modeling of (2) NarH was generated based on sequence residues from 1-505 with 99% coverage, 100% confidence, and 71% identity against D1Y5ib1 (1.90 Å). (3) NarI was generated based on sequence residues from 6-173 with 86% coverage, 100% confidence, and 50% identity against D1Y5ic1 (1.90 Å). Furthermore, other nitrogen metabolic proteins (NrtP, NasA, NarZ, and NxrA), nitrous-oxide reductase (NosZ) [EC:1.7.2.4], glutamine synthetase (GlnA and GluL) [EC:6.3.1.2], glutamate dehydrogenase (GudB and RocG) [EC:1.4.1.2], glutamate synthase (NADPH) large chain (GltB) [EC:1.4.1.13], carbamate kinase (CK) [EC:2.7.2.2], and carbonic anhydrase (CA) [EC:4.2.1.1] were identified. PTS IIA-like nitrogen-regulatory protein (PtsN), two-component nitrogen fixation transcription regulator (FixJ), and nitrate/nitrite transporter (NRT2) were also identified. Although the *Mesorhizobium* PAMC28654 strain showed some nitrogen metabolic enzymes such as Nar, it lacked further denitrification proteins. This coincided with many microorganisms well reported to be able to reduce nitrate to nitrite without further denitrification enzymes [38,39].

When nitrogen metabolic enzymes of complete genomes of all *Mesorhizobium* strains were compared using blastKOALA, the nitrilase enzyme that could convert nitriles to carboxylic acids and ammonia through hydrolysis without forming free amide intermediates [40] was identified in *Mesorhizobium* sp. L-8-10, *Mesorhizobium* sp. L-8-3, *Mesorhizobium* sp. M1B.F.Ca.ET.045.04.1.1, *Mesorhizobium* sp. M1D.F.Ca.ET.043.01.1.1, two strains of *Mesorhizobium* sp. NZP2077, *Mesorhizobium* sp. Pch-S, and *Mesorhizobium* loti R88. However, the remaining strains including our strain lacked the nitrilase enzyme (Appendix A). In addition, complete genomes of most *Mesorhizobium* (Appendix A) showed the presence of a nitrogen fixation (Nif) enzyme that could convert atmospheric nitrogen to ammonia and other related nitrogenous compounds. Most of the studies have investigated nitrogen fixation and nodulation. However, Nif enzymes were not identified in *Mesorhizobium* sp. 8, *Mesorhizobium* sp. B1-1-8, *Mesorhizobium* sp. B2-1-1, *Mesorhizobium* sp. B2-1-8, *Mesorhizobium* sp. B2-8-5, *Mesorhizobium* sp. B4-1-4, *Mesorhizobium* sp. INR15, *Mesorhizobium* sp. NBSH29, *Mesorhizobium* sp. NZP2077, and *Mesorhizobium* sp. Pch-S, including our strain.

From genome analysis, a nitrate reductase enzyme was identified. Thus, the strain was subjected to a nitrate reduction assay. For this nitrate reduction assay, the isolated strain (*Mesorhizobium* sp. PAMC28654), an abiotic control (without any microorganism), and a positive control (with *E. coli* strain) were used. A change in color from a colorless solution to red was monitored. The *E. coli* strain and our isolated strain showed a change in color from colorless to red at both temperatures of 15 °C and 25 °C, confirming the reduction in nitrate (Appendix A). However, the abiotic control did not show any color change.

### 3.5. EPS Production and SEM

Genome analysis of the isolated strain showed EPS-producing proteins ExoF and ExoQ. The strain demonstrated EPS production at both temperatures of 15 °C and 25 °C (Appendix A). However, the type of EPS produced (slime or capsular) needs further characterization. The weight of the crude-EPS produced was 0.115 g/100 mL at 15 °C and 0.075 g/100 mL at 25 °C after drying. The weight of EPS was a little bit higher at 15 °C than at 25 °C, consistent with a study conducted by Ali et al. [41] on *Pseudomonas* sp. Bgl2 where the yield of cryoprotective EPS was higher at a lower temperature. In addition, the surface morphology of extracted dried EPS by SEM was analyzed (Appendix A). EPSs isolated from *Mesorhizobium* sp. PAMC28654 might have other applications such as for environmental, industrial, and agricultural purposes. This needs a further extensive study.

### 3.6. Salinity Test

Salinity tests of the isolated bacterial strain were performed at two different temperatures of 15 °C and 25 °C using different concentrations (500 mM, 100 mM, 50 mM, 10 mM, 5 mM, 2.5 mM, 2 mM, 1 mM, 0.5 mM, 0.25 mM, 0.1 mM, 0.05 mM, and 0.01 mM) of NaCl (Figure 5). A negative control without any bacteria and a positive control with bacteria were also included in this test. *Mesorhizobium* sp. PAMC28654 showed tolerance for NaCl at concentrations up to 100 mM at both temperatures of 15 °C and 25 °C. However, no difference in salt tolerance concentration was observed at both temperatures. Our result agrees with the reported literature that most of the rhizobia strain is inhibited by 100 mM NaCl while some can tolerate more than 300 mM. The majority of the work related to salinity has been performed on *Mesorhizobium* isolated from chickpeas. The salinity concentrations measured varied from 0 to 1.71 mM to 170 mM to 1034 mM [42,43]. When *Mesorhizobium* strains (Appendix A) are compared, *Mesorhizobium* loti MAFF303099 has been reported to be salt-sensitive (28% growth with 1.71 mM of NaCl) [42]. Some *Mesorhizobium* species isolated from acacias in arid and semi-arid regions in Algeria have shown variabilities in their tolerance for NaCl. The salinity tolerance of our strain was higher than *Mesorhizobium* loti MAFF303099 but less than other *Mesorhizobium* species isolated from Algeria. Salt tolerance and dependence are characteristics of some microorganisms with the potential to tolerate salt stress. These microorganisms could be used for agricultural purposes to overcome and manage the detrimental effects of saline soil for better crop productivity [16,44]. Although our bacterial strain did not show differences in salt tolerance at two different temperatures, this strain did show a capacity to tolerate NaCl at concentrations up to 100 mM. Thus, this strain might have the potential to be used for agricultural purposes in the future. This needs further study.

### 3.7. Toxicity of Trace Elements

Microbes confer various types of resistance mechanisms in response to HMs [45]. Efflux pumps are major known players in resistance mechanisms with both plasmid and chromosomal systems [46]. Copper-resistant protein CopC, CopD, arsenical-resistant protein Acr3, cobalt, zinc, and cadmium-resistant protein were identified from genome analysis of the isolated strain (Table 2). Thus, only those trace elements were considered and subjected to wet-lab experiments. For wet-lab experiments, *Mesorhizobium* sp. PAMC28654 cells were used at an initial concentration of ~2.42 × 10^8^ cells/mL to measure HMs toxicity at two different temperatures of 15 °C and 25 °C. Among five trace elements (CuSO_4_·5H_2_O, CoCl_2_·6H_2_O, ZnSO_4_·7H_2_O, Cd(NO_3_)_2_·4H_2_O, and Na_2_HAsO_4_·7H_2_O) tested, it was found that the *Mesorhizobium* sp. PAMC28654 strain could tolerate 1 mM CuSO_4_·5H_2_O, 2 mM CoCl_2_·6H_2_O, 1 mM ZnSO_4_·7H_2_O, 0.05 mM Cd(NO_3_)_2_·4H_2_O, and 100 mM Na_2_HAsO_4_·7H_2_O at 15 °C and 0.25 mM CuSO_4_·5H_2_O, 2 mM CoCl_2_·6H_2_O, 0.5 mM ZnSO_4_·7H_2_O, 0.01 mM Cd(NO_3_)_2_·4H_2_O, and 100 mM Na_2_HAsO_4_·7H_2_O at 25 °C (Figure 6). These results demonstrated that *Mesorhizobium* sp. PAMC28654 had different trace element tolerance at two different temperatures of 15 °C and 25 °C. The cold temperature (15 °C) was more favorable for the isolated strain’s tolerance to different trace elements. Comparing *Mesorhizobium* strains with complete genomes from Appendix A and draft genomes of *Mesorhizobium* reported in the NCBI database and literature so far, HM-related studies on *Mesorhizobium* species have been mostly conducted on copper, zinc, lead, cadmium, and chromium [14,47,48]. The resistance capacity of the *Mesorhizobium* PAMC28654 strain for trace elements such as zinc and cadmium was lower than a strain of *Mesorhizobium metallidurans* belonging to the same genus isolated from the metallicolous soil of a zinc mining area in Languedoc, France, reported by Vidal et al. [14]. They found that their strain could tolerate high concentrations of HMs (16–32 mM of Zn and 0.3–0.5 mM of Cd) at 28 °C. Although our isolated strain showed less resistance to trace elements than *Mesorhizobium metallidurans,* the fact that polar areas are not exposed to highly contaminated areas like in the case of *Mesorhizobium metallidurans* should be considered. HM, particularly arsenic, has been studied very little in *Mesorhizobium* sp. The findings in this study will be valuable in terms of studying the diversity of arsenic-resistant bacteria. The *Mesorhizobium* PAMC28654 strain might have the potential for phytoremediation and bioremediation of HMs similar to some HM-resistant bacteria such as *Mesorhizobium* sp. *loti* HZ76 and *Mesorhizobium cicero*, both of which have demonstrated the ability to improve phytoremediation [47,48]. For that purpose, these preliminary results need further extensive studies.

## 4. Conclusions

In this study, we isolated a *Mesorhizobium* strain, namely *Mesorhizobium* sp. PAMC28654, from a polar region of Uganda. The strain demonstrated abilities to produce EPSs and reduce nitrate. It also showed salt tolerance (NaCl) at concentrations up to 100 mM at both temperatures of 15 °C and 25 °C and tolerance to multiple trace elements. Among the five trace elements (CuSO_4_·5H_2_O, CoCl_2_·6H_2_O, ZnSO_4_·7H_2_O, Cd(NO_3_)_2_·4H_2_O, and Na_2_HAsO_4_·7H_2_O) tested at two different temperatures of 15 °C and 25 °C, the isolated strain could tolerate 1 mM CuSO_4_·5H_2_O, 2 mM CoCl_2_·6H_2_O, 1 mM ZnSO_4_·7H_2_O, 0.05 mM Cd(NO_3_)_2_·4H_2_O, and 100 mM Na_2_HAsO_4_·7H_2_O at 15 °C and 0.25 mM CuSO_4_·5H_2_O, 2 mM CoCl_2_·6H_2_O, 0.5 mM ZnSO_4_·7H_2_O, 0.01 mM Cd(NO_3_)_2_·4H_2_O, and 100 mM Na_2_HAsO_4_·7H_2_O at 25 °C. The adaptation of a bacterial strain towards different stresses such as cold temperatures, high salt, and toxic trace elements will be very valuable in terms of understanding bacteria from polar regions. Furthermore, these studies are helpful for exploring the diversity of trace element-resistant microorganisms, understanding the overlapping gap between biotic and abiotic processes, and monitoring environmental health and can be implemented for environmental and agricultural purposes in the future.

## Figures and Tables

**Figure 1 microorganisms-12-00120-f001:**
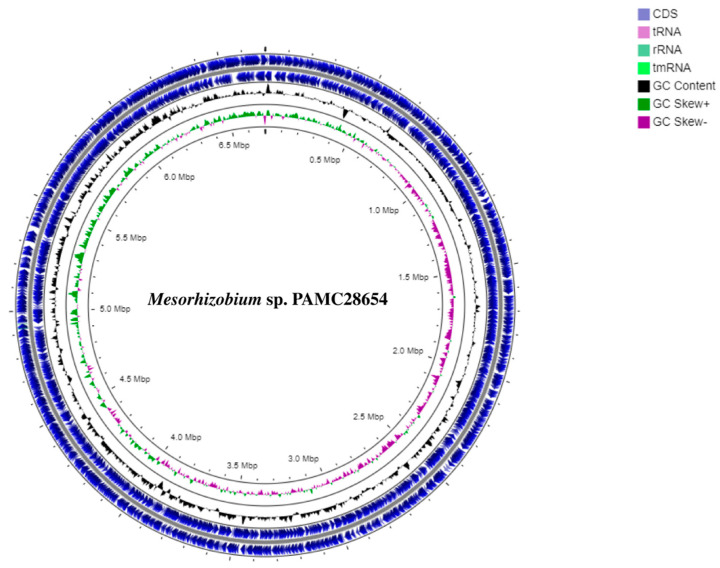
Circular map and total genome information of *Mesorhizobium* sp. PAMC28654. Outer circle to inner circle: CDS, blast, GC Content, GC Skew+, GC Skew−, and GC.

**Figure 2 microorganisms-12-00120-f002:**
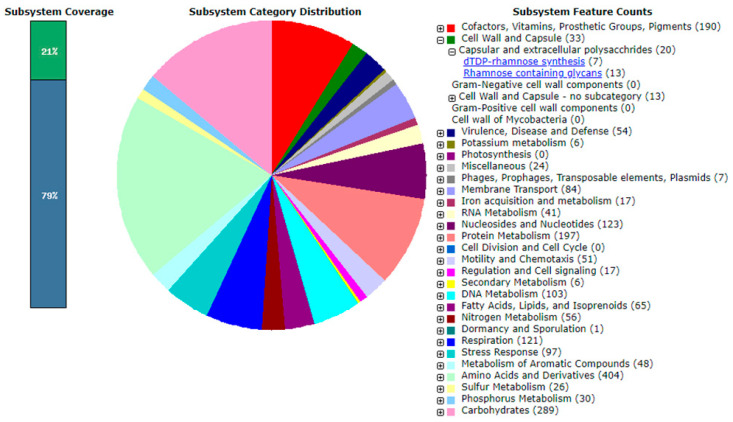
Subsystem distribution of *Mesorhizobium* sp. PAMC28654. The number in parentheses is the number of protein-coding genes predicted to be involved in that cellular process. The underlined text indicates subsystem features involved in dTDP-rhamnose synthesis and rhamnose-containing glycans.

**Figure 3 microorganisms-12-00120-f003:**
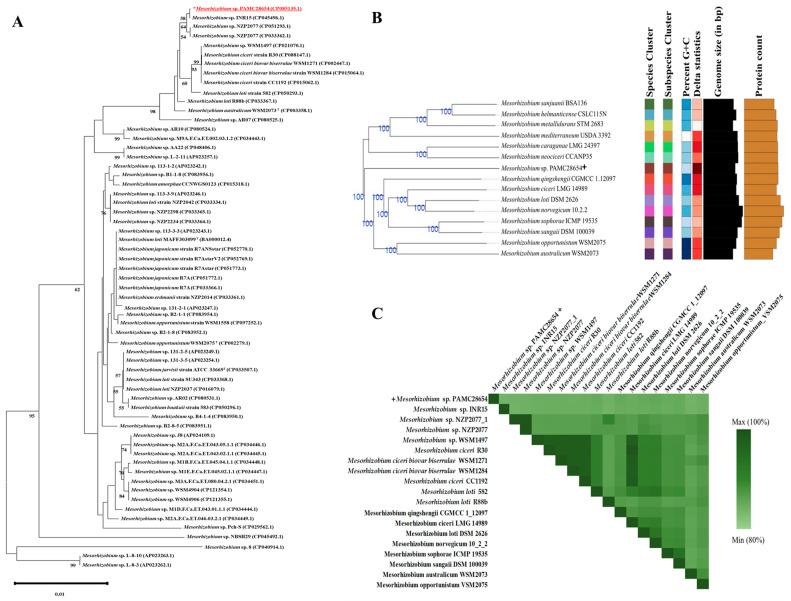
(**A**) Neighbor-joining phylogenetic trees of 16S rRNA gene of *Mesorhizobium* sp. PAMC28654 with other *Mesorhizobium* strains without an outgroup using maximum composite likelihood model. The size of the alignment of 16S rRNA sequences is 1496 nucleotides (approximately 1500 nucleotides). Percentages in the bootstrap test are from 1000 sample replicates. Only values above 50% are shown in branch nodes. “*” denotes *Mesorhizobium* sp. PAMC28654 strain (**B**) TYGS results for whole-genome sequence of *Mesorhizobium* datasets. “_+_” denotes *Mesorhizobium* sp. PAMC28654 strain. Different colors are provided to indicate species and subspecies clusters. Same color denotes the same species cluster (**C**) Orthologous Average Nucleotide Identity (ANI) results of our strain *Mesorhizobium* sp. PAMC28654 with other selected *Mesorhizobium* strains available in the NCBI database. “_+_” denotes *Mesorhizobium* sp. PAMC28654 strain.

**Figure 4 microorganisms-12-00120-f004:**
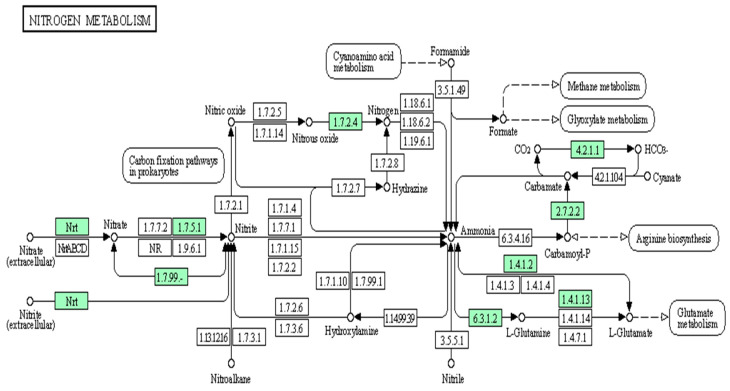
Snapshot of KEGG database showing nitrogen metabolism pathway in *Mesorhizobium* sp. PAMC28654. Enzymes available are highlighted in green.

**Figure 5 microorganisms-12-00120-f005:**
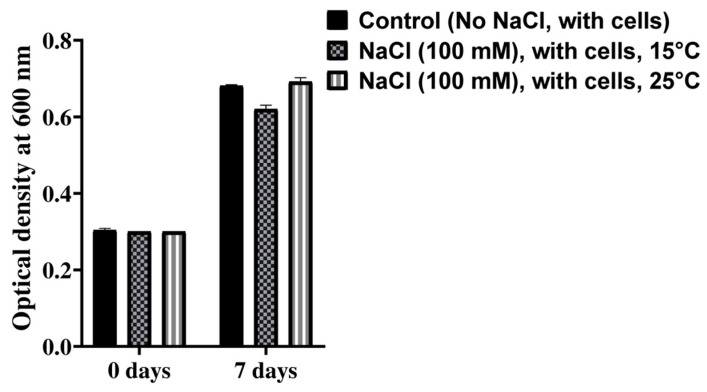
Salinity test of *Mesorhizobium* sp. PAMC28654 strain at two different temperatures of 15 °C and 25 °C. Black color represents control (No NaCl, with *Mesorhizobium* sp. PAMC28654 cells) and different filled pattern represents test (NaCl, with *Mesorhizobium* sp. PAMC28654 cells at 15 °C and 25 °C).

**Figure 6 microorganisms-12-00120-f006:**
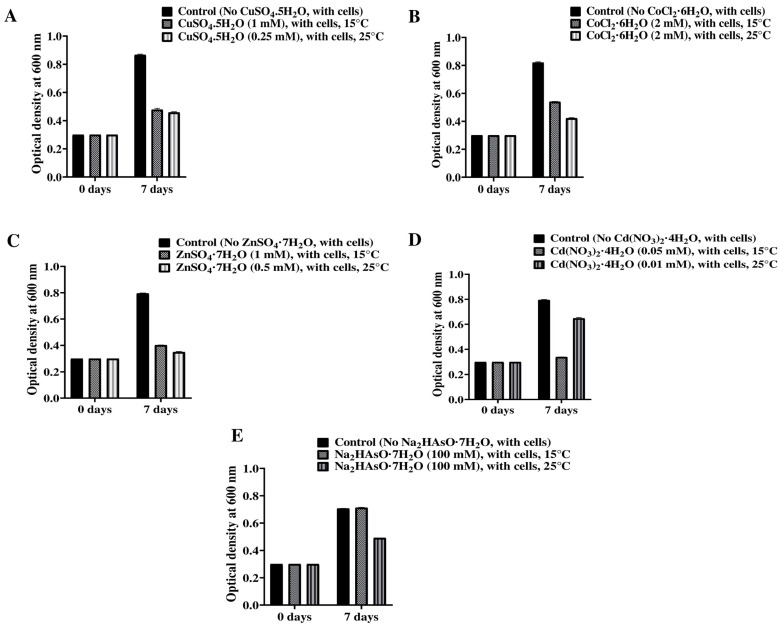
Effects of trace elements on growth of *Mesorhizobium* sp. (**A**–**E**) PAMC28654 strain at 15 °C (left panel) and 25 °C (right panel). Black color represents control (no trace elements, with *Mesorhizobium* sp. PAMC28654 cells) and different filled pattern represents test (trace elements, with *Mesorhizobium* sp. PAMC28654 cells at 15 °C and 25 °C).

**Table 1 microorganisms-12-00120-t001:** Genome features of *Mesorhizobium* sp. PAMC28654.

Feature	Value
**Genome statistics**	
Contigs	1
Total length (bp)	6,701,426
N_50_	6,701,426
L_50_	1
GC (%)	62.2
**Genome features**	
Assembly level	Complete genome
Chromosome genes	6642
Protein-coding genes	6178
Pseudogenes	400
rRNA genes	6
tRNA genes	54

**Table 2 microorganisms-12-00120-t002:** Resistant protein, transcriptional regulator, and transporter in the genome of *Mesorhizobium* sp. PAMC28654.

Enzyme/Transcription Factor/Transporter
Cobalt-zinc-cadmium resistance protein CzcD
Lead, cadmium, zinc, and mercury transporting ATPase
Copper-translocating P-type ATPase
Heavy metal resistance transcriptional regulator HmrR
Zinc uptake regulation protein Zur
Copper resistance protein CopC
Copper resistance protein CopD
Arsenical resistance operon repressor
Arsenical resistance protein Acr3
Arsenate reductase glutaredoxin-coupled, glutaredoxin-like family
Transcriptional regulator, ArsR family
Magnesium and cobalt efflux protein CorC
Magnesium and cobalt transport protein CorA

## Data Availability

Data are contained within the article or Appendix A. Complete genome sequences of *Mesorhizobium* sp. PAMC28654 were deposited in the National Center for Biotechnology Information (NCBI) GenBank database. The Genbank accession numbers are PRJNA768962 for the BioProject, SAMN22072446 for the BioSample. Datasets analyzed during the current study are available in the NCBI repository with the following accession numbers: NZ_CP085135.1 for *Mesorhizobium* sp. PAMC28654, NZ_AP023242.1 for *Mesorhizobium* sp. 113-1-2, NZ_AP023243.1 for *Mesorhizobium* sp. 113-3-3, NZ_AP023246.1 for *Mesorhizobium* sp. 113-3-9, NZ_AP023247.1 for *Mesorhizobium* sp. 131-2-1, NZ_AP023249.1 for *Mesorhizobium* sp. 131-2-5, NZ_AP023254.1 for *Mesorhizobium* sp. 131-3-5, NZ_CP040914.1 for *Mesorhizobium* sp. 8, NZ_CP048406.1 for *Mesorhizobium* sp. AA22, NZ_CP080531.1 for *Mesorhizobium* sp. AR02, NZ_CP080525.1 for *Mesorhizobium* sp. AR07, NZ_CP080524.1 for *Mesorhizobium* sp. AR10, NZ_CP083956.1 for *Mesorhizobium* sp. B1-1-8, NZ_CP083954.1 for *Mesorhizobium* sp. B2-1-1, NZ_CP083952.1 for *Mesorhizobium* sp. B2-1-8, NZ_CP083951.1 for *Mesorhizobium* sp. B2-8-5, NZ_CP083950.1 for *Mesorhizobium* sp. B4-1-4, NZ_CP045496.1 for *Mesorhizobium* sp. INR15, NZ_ AP024109.1 for *Mesorhizobium* sp. J8, NZ_AP023257.1 for *Mesorhizobium* sp. L-2-11, NZ_AP023263.1 for *Mesorhizobium* sp. L-8-10, NZ_AP023262.1 for *Mesorhizobium* sp. L-8-3, NZ_CP034448.1 for *Mesorhizobium* sp. M1B.F.Ca.ET.045.04.1.1, NZ_CP034444.1 for *Mesorhizobium* sp. M1D.F.Ca.ET.043.01.1.1, NZ_CP034447.1 for *Mesorhizobium* sp. M1E.F.Ca.ET.045.02.1.1, NZ_CP034445.1 for *Mesorhizobium* sp. M2A.F.Ca.ET.043.02.1.1, NZ_CP034446.1 for *Mesorhizobium* sp. M2A.F.Ca.ET.043.05.1.1, NZ_CP034449.1 for *Mesorhizobium* sp. M2A.F.Ca.ET.046.03.2.1, NZ_CP034451.1 for *Mesorhizobium* sp. M3A.F.Ca.ET.080.04.2.1, NZ_CP034443.1 for *Mesorhizobium* sp. M9A.F.Ca.ET.002.03.1.2, NZ_CP045492.1 for *Mesorhizobium* sp. NBSH29, NZ_CP051293.1 for *Mesorhizobium* sp. NZP2077, NZ_CP033362.1 for *Mesorhizobium* sp. NZP2077, NZ_CP033364.1 for *Mesorhizobium* sp. NZP2234, NZ_CP033365.1 for *Mesorhizobium* sp. NZP2298, NZ_CP029562.1 for *Mesorhizobium* sp. Pch-S, NZ_CP021070.1 for *Mesorhizobium* sp. WSM1497, NZ_CP121354.1 for *Mesorhizobium* sp. WSM4904, NZ_CP121355.1 for *Mesorhizobium* sp. WSM4906, NZ_CP088147.1 for *Mesorhizobium ciceri* strain R30, NZ_CP015062.1 for *Mesorhizobium ciceri* strain CC1192, NZ_CP015064.1 for *Mesorhizobium ciceri biovar biserrulae* strain WSM1284, NZ_CP002447.1 for *Mesorhizobium ciceri biovar biserrulae* WSM1271, NZ_CP033368.1 for *Mesorhizobium loti* strain SU343, NZ_CP033334.1 for *Mesorhizobium loti* strain NZP2042, NZ_CP050293.1 for *Mesorhizobium loti* strain 582, NZ_CP016079.1 for *Mesorhizobium loti* NZP2037, NZ_CP033367.1 for *Mesorhizobium loti* R88b, NZ_CP051773.1 for *Mesorhizobium japonicum* strain R7Astar, NZ_CP052769.1 for *Mesorhizobium japonicum* strain R7AstarV2, NZ_CP052770.1 for *Mesorhizobium japonicum* strain R7ANSstar, NZ_BA000012.4 for *Mesorhizobium loti* MAFF303099, NZ_CP051772.1 for *Mesorhizobium japonicum* R7A, NZ_CP033366.1 for *Mesorhizobium japonicum* R7A, NZ_CP015318.1 for *Mesorhizobium amorphae* CCNWGS0123, NZ_CP050296.1 for *Mesorhizobium huakuii* strain 583, NZ_CP003358.1 for *Mesorhizobium australicum* WSM2073, NZ_CP097252.1 for *Mesorhizobium opportunistum* strain WSM1558, NZ_CP002279.1 for *Mesorhizobium opportunistum* WSM2075, NZ_CP033361.1 for *Mesorhizobium erdmanii* strain NZP2014, NZ_CP033507.1 for *Mesorhizobium jarvisii* strain ATCC 33669.

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
