# Peer review of "Comparative Genome Analysis of Polar Mesorhizobium sp. PAMC28654 to Gain Insight into Tolerance to Salinity and Trace Element Stress"

_microorganisms, 2024, doi:10.3390/microorganisms12010120_

Round 1

Reviewer 1 Report

Comments and Suggestions for Authors

“Comparative genome analysis of polar Mesorhizobium sp. PAMC28654 to gain insight into tolerance for salinity and heavy metals stress”.

The manuscript contributes with a complete genome sequence of the strain Mesorhizobium PAMC28654 isolated from soils of Rwenzori Mountains in Uganda and proposes a genomic comparative study including 61 strains with available genome sequence in NCBI. Several annotation programs were used to accurately define the coding sequences related to environmental stresses of the PAMC28654. The genomic comparison was based on 16S rRNA phylogeny (Figure 3), statistics of the genome sequences (Tables 2 and 3), and genome annotation. In addition, the temperature, salt, and heavy metals tolerance, as well as nitrate reduction and EPS production of the strain PAMC28654 were evaluated. The main findings refer to the isolation of a Mesorhizobium strain from cold regions, an uncommon report in the literature, and the salt and heavy metals tolerance of this strain. The manuscript contains supplementary material; however, it is not available to download.

Several tools and approaches may be applied to qualify a study as comparative genomics. Therefore, I suggest performing a robust core genome phylogeny based on phylogenomic analysis to demonstrate the relatedness of the strains; as well as better exploring the genome annotation of the closest strains and PAMC28654 in light of the features of interest trying to evidence the presence/absence of genes related to tolerance to stressful conditions.

Since the document with line numbers is not available, my comments are described here followed by the copied sentences from the manuscript, which were highlighted in the PDF document.

[Comment 1]:

Introduction section:

[Comment 1]: “The Mesorhizobium genus was identified by Jarvis et al., in 1997 [1] is a member of the family Phyllobacteriaceae, phylum Pseudomonadota, and class Alphaproteobacteria [2,3].”

Add an "and" before "is a member".

[Comment 2]: “The main aim of the study was to isolate a new Mesorhizobium species from cold re-gions and study their possibilities for their survival in cold areas.”

The analysis provided in the manuscript did not allow to affirm that strain Mesorhizobium PAMC28654 is a new species, therefore the term “a new Mesorhizobium species” must be replaced with “a Mesorhizobium strain”.

Materials and methods section:

[Comment 3]: “The strain Mesorhizobium sp. PAMC28654 was isolated from soil samples collected in the Rwenzori Mountains of Uganda by the Korea Polar Research Institute (KOPRI, In-cheon, Korea).”

Please provide the geographic coordinates of the local where the strain was collected as well as more information about the isolation process (how the soil was collected, bacteria isolation method, culture media, growth conditions, pure colonies identification, etc).

[Comment 4]: “Similarly, we obtained 61 genomes of the Mesorhizobium species from the NCBI data-base.”

-        Several genomes from NCBI used in the study are from Mesorhizobium strains only classified to genus level, therefore the term “61 genomes of the Mesorhizobium species” must be replaced with “61 genomes of Mesorhizobium strains.” ;

-        Specify how the 61 genomic sequences were selected.

[Comment 5]: “Phylogenetic tree was constructed using 16S rRNA sequences of the complete ge-nomes of uncategorized and categorized Mesorhizobium sp. obtained from NCBI 16S rRNA were aligned using MUSCLE and MEGA 11 [22–27]”.

-        Please replace “Mesorhizobium sp.” with “Mesorhizobium strains”;

-        Provide the phylogenetic method and evolutionary model used to construct the tree and the number of re-samplings for bootstrap analysis.

Results and discussion section

[Comment 6]: Table 2 and Table 3

-        Since not all genomes from strains used in the manuscript are classified as a species, delete the “sp.” between “Mesorhizobium” and “strains” of the legend of both tables;

-        Allocate Tables 2 and 3 in the supplementary material;

-        Add the meaning of the N/A abbreviation in the footnotes of both tables;

-        In the column “Host” of Table 2, change the species names to italic;

-        Add a column with the accession number of genome sequences used in Table 2 of the manuscript;

-        In the column “Protein count” of Table 3, put the total number of proteins;

-        Indicate if the strain used is a type strain of a species by adding the superscript T (T) in the columns “Strain” and “Strain name” in Tables 2 and 3, respectively;

-        Since not all genomes from strains used in the manuscript are classified as a species, replace “Mesorhizobium sp. strains” with “Mesorhizobium strains” from the legend of both tables.

[Comment 7]: “Phylogenetic tree was constructed using 16S rRNA sequences of the complete ge-nomes of uncategorized and categorized Mesorhizobium sp. obtained from NCBI by using MEGA 11 (https://www.megasoftware.net/) showed that our strain is most closely re-lated to Mesorhizobium sp. INR15 (similarity of 99.87%, CP045496.1) […]”

Replace “Mesorhizobium sp.” with “Mesorhizobium strains”. The abbreviation sp. refers to a single species and spp. for several species. However, since not all genomes from strains used in the manuscript are classified as species, it must be properly fixed in the rest of the manuscript.

[Comment 8]: Phylogenetic comparison

Even though the 16S rRNA phylogeny is a technique broadly used in bacterial taxonomy, it provides information limited to genus level identification. It is hard to infer the relatedness of strains in a genus based uniquely on 16S rRNA phylogeny because the sequence of this gene is highly conserved. Since the manuscript proposes a comparative genomic study, and the authors selected a great number of complete genomic sequences of Mesorhizobium strains available in NCBI, I suggest constructing a robust phylogeny based on the core genome of the strains.

[Comment 9]: In the legend of Figure 3, add the size of the alignment of 16S rRNA sequences.

[Comment 10]: “Putative 3D structure modelling of the all three subunit Nar enzymes (NarG, α-subunit; NarH, β subunit; and NarI, γ subunit) were shown in (Supplementary Fig. S1)” and “The change in color to red from colorless solution was monitored. E. coli strain and our isolated strain showed change in color from colorless to red at both the temperatures 15°C and 25°C, confirming the reduc-tion of nitrate whereas abiotic control did not show any color change (Supplementary Fig. S2).”

Supplementary material is not available to download.

[Comment 11]: “Furthermore, other nitrogen met-abolic proteins (nrtP, nasA, narZ, and nxrA), nitrous-oxide reductase (nosZ) [EC:1.7.2.4], glutamine synthetase (glnA and glul) [EC:6.3.1.2], glutamate dehydrogenase (gudB and rocG) [EC:1.4.1.2], glutamate synthase (NADPH) large chain (gltB) [EC:1.4.1.13], carba-mate kinase (arc) [EC:2.7.2.2], and carbonic anhydrase (CA) [EC:4.2.1.1], were identi-fied.”

The protein abbreviation must be written with the first letter capitalized and not italic (e.g. NrtP), whereas the gene abbreviation must be written in lowercase and italic (e.g. nrtP - italic). Please fix the nomenclature of proteins/genes properly in the rest of the manuscript.

[Comment 12]: “Comparing the nitrogen metabolic enzymes of the complete genome of all the Meso-rhizobium sp., nitrilase enzyme that convert the hydrolysis of nitriles to carboxylic acids and ammonia without the formation of free amide intermediates [45] were identified in Mesorhizobium sp. L-8-10, Mesorhizobium sp. L-8-3, Mesorhizobium sp. M1B.F.Ca.ET.045.04.1.1, Mesorhizobiumsp. M1D.F.Ca.ET.043.01.1.1, two strains of Mesorhi-zobium sp. NZP2077, Mesorhizobium sp. Pch-S, and Mesorhizobium loti R88 but in remaining strain including our strain lack the nitrilase enzyme (Table 2)”.

It is not clear if all genomes retrieved from NCBI were annotated with the same programs as PAMC28654.

[Comment 13]: “Comparing the Mesorhizobium sps from Table 2. in-cluding draft genome of Mesorhizobium reported in the NCBI database so far HMs related work in the Mesorhizobium sps has been mostly done in copper, zinc, lead, cadmium, and chromium”.

The sentence is confusing, please rewrite it. Are the 61 genomic sequences draft or finished?

[Comment 14]: “In this study, we have isolated a new Mesorhizobium species, namely Mesorhizobium sp. PAMC28654 from the polar region of Uganda.”

Based on the results presented in the manuscript, it is not possible to affirm that PAMC28654 is a new species. 

Comments on the Quality of English Language

The manuscript presents some grammar mistakes, and it should be carefully reviewed. Please review the English in the following sentences: 

“The broad distribution of this genus and its ability to make a symbiotic relationship to several plant genera makes it an interesting candidate for agronomic and ecological pro-spective [7,8].”

Replace “to” with “with”.

“In addition to that, use of EPS produced by rhizobium such as Mesorhizo-bium has contributed significantly to industrial purposes as their use as a gelling, thicken-ing, and stabilizing agents in foods, pharmaceutical, and cosmetics [9].”

-        Add a “the” before use.

-        Replace “pharmaceutical” with “pharmaceuticals”.

“Furthermore, abiotic stresses such as heavy metals (HMs), tempera-ture stress, salinity, nutrient availability stress has been reported in the polar areas. “

-        Add an “and” between salinity and nutrient availability.

-        Replace “has” with “have”.

“Bacteria has been reported to tolerate a wide range of temperatures from extreme high temperature (thermophiles) to extreme low temperature (psychrophiles). Bacteria mediate the ability to adjust to low or cold temperature by structural adjustment of enzymes, maintenance of membrane fluidity, expression of cold shock protein, and adaptation of the translation and transcription machinery [10,12]. “

-        Replace “has” with “have”.

-        Replace the two “extreme” with “extremely”.

-        Replace “temperature” with “temperatures”.

“Next to temperature, HMs (those elements having high atomic weight and atomic number such as copper, Cu; zinc, Zn; cadmium, Cd; arse-nate, As; lead, Pb; mercury, Hg; chromium, Cr; and nickel, Ni) stress is another factor that has huge influence on bacterial survival.”

Replace “has huge” with “have a huge”.

“Be-sides temperature and HMs stress, salt stress is yet another factor that have huge impact on the bacterial growth and survival, and biggest hurdle in achieving better crop yield and quality with soil having high salinity [16]. “

-        Replace “have huge” with “has a huge”.

-        Add a “the” before biggest.

“Even though, Mesorhizobium species has been studied mostly at the optimum temperature (25°C–30°C) [17] for agronomic, ecological, and industrial applica-tion [18], and very limited studies has been carried out for extreme environmental condi-tion as low temperature particularly for issues like HMs and salinity. “

Replace “has been studied” with “have been studied” and “has been carried” with “have been carried”.

“This study is very helpful to a better understanding of bacteria mediating their survival on cold temperature as well as other stress, such as salinity and HMs toxicity through the production of EPS. In addition to that, the isolated strain might be implemented in the future for environmental and agricultural aspects.”

Replace “cold temperature” with “cold temperatures”.

The isolated strains were checked for their growth in several media, including R2A, lysogeny agar (LBA), nutrient agar (NA), tryptic soy agar (TSA), and marine agar (MA) at three different temperature 15°C, 25°C, and 37°C. “

Replace “different temperature” with “different temperatures”.

“The putative 3D structure of nitrate reductase was modeled by online program PHYRE2 Server (http://genome3d.eu/) in the intensive mode based on the most identical template in the PDB database [28].”

Add a “the” before online program.

“For the confirmation, a pinch of zinc dust was added to the tube with reagent A and B. “

Replace “reagent” with “reagents”.

“The isolated strain was tested for EPS production at both temperature (15°C and 25°C). “

Replace “temperature” with “temperatures” in the sentence, as well as in the rest of the manuscript when it is describing two or more temperatures.

“Negative control without the bacteria and positive control using the bacteria were also used. The bacterial growth OD600 was measured by spectrophotometer. Initial concentration of bacterial cells used for the salinity toxicity was ˜2.42 × 108 cells/ml (reference E. coli cells, OD600 of 1.0 = 8 × 108 cells/ml). “

Add an “a” before spectrophotometer and a “The” before initial.

“Initial con-centration of bacterial cells used for the HMs toxicity was ˜2.42 × 108 cells/ml (reference E. coli cells, OD600 of 1.0 = 8 × 108 cells/ml).”

Add a “The” before initial.

“The isolated strains Mesorhizobium sp. PAMC28654 showed its growth in R2A broth and nutrient agar (NA) at two different temperature 15°C and 25°C, besides that, the strain showed their growth at specific media such as in the nitrate broth and the nitrite broth at 15°C and 25°C. “

Replace “strains” with “strain” and “temperature” with “temperatures”.

“Nar enzymes have been reported to be responsible for the conversion of initial step in the nitrate reduction pathway and often in anaerobic or low-oxygen condi-tion. “

Add a “the” before initial.

“NarG is responsible for binding and reducing nitrate to nitrite, which facilitates the reduction of nitrate by transferring the electrons to NarG, and NarI, anchor the NarG and NarH subunit to the bacterial membrane.”

Replace “anchor” with “anchoring”.

“ In addition to that, nitrate/nitrite protein (Nark), responsible for importing nitrate from the extracellular environment into the bacterial cell to be utilized by the nitrate reductase complex.”

Remove the comma after (NarK) and add “is” before responsible.

“Putative 3D structure modelling of the all three subunit Nar enzymes (NarG, α-subunit; NarH, β subunit; and NarI, γ subunit) were shown in (Supplementary Fig. S1). “

Remove “all” before three.

“Even though, our strain showed some nitrogen metabolic enzymes such as Nar but lack of further denitrification proteins which coincide with many microorganisms that has been well reported to reduce nitrate to nitrite but lack further denitrification enzymes [43,44]. “

-        Replace “lack of” with “lacked”.

-        Replace “has been well reported” with “have been well reported”.

“[…] but in remaining strain including our strain lack the nitrilase enzyme (Table 2).”

Replace “strain” with “strains”.

“The change in color to red from colorless solution was monitored. E. coli strain and our isolated strain showed change in color from colorless to red at both the temperatures 15°C and 25°C, confirming the reduc-tion of nitrate whereas abiotic control did not show any color change (Supplementary Fig. S2).”

Add an “a” before change in color.

The genome analysis of the isolated strain showed EPS-producing protein ExoF and ExoQ. “

Replace “protein” with “proteins”.

“The weight of EPS was little bit higher at 15°C than at 25°C, which complies with the study done by Perviaz ali et al., [46] on Pseu-domonas sp. “

Add a “a” before little.

“The strain showed tolerance of NaCl up to 100 mM at both the temperature at 15°C to 25°C. However, no difference in salt tolerance concentration was observed at both tem-peratures. “

Replace “both the temperature” with “both temperatures”.

“In addition to that, some Mesorhizobium sps isolated from acacias in arid and semi-arid regions in Al-geria has shown to variability in the tolerance of NaCl. “

Replace “has” with “have” and remove “to” before variability.

“Salt tolerance and dependence are the characteristics of some micro-organism and microorganism-having potential to tolerate salt stress therefore could be implemented for agricultural purpose to overcome detrimental effect and management of saline soil for better crop productivity [16,49] considering that, even though our bacterial strain did not show differences in salt tolerant at two different temperatures, but still the strain has the capacity to tolerate NaCl up to 100 mM so the strain might have the potential to implemented for agricultural purpose in future needs further study.”

-        Put “therefore” between commas, replace “purpose” with “purposes”, “effect” with “effects” and “tolerant” with “tolerance”.

-        Add a “to” between “to” and “implemented” and replace “purpose” with “purposes”.

“Microbes confer various types of resistance mechanism in response to HMs [50] and efflux pumps are the major known players for group of resistance systems with both plas-mid and chromosomal system [51]. “

Put “mechanism” and “system” in the plural.

“HM resistant protein such as copper resistant protein CopC, CopD, arsenical resistant protein ACR3, cobalt, zinc, and cadmium resistant pro-tein were identified from the genome analysis (Table 4) of the isolated strain so, only those HMs were considered and subjected to the wet-lab experiments. “

Replace “HM resistant protein” with “HM resistant proteins”.

“The result demonstrated difference in different HMs tolerance ability of our isolated strain at two different temper-atures (15°C and 25°C). “

Replace “difference” with “differences”.

“The resistance capacity of our strain for HMs such as zinc and cad-mium are less as compared to the study done by Celine Vidal et al., [14] on Mesorhizobium metallidurans belonging to the same genus isolated from the metallicolous soil of zinc min-ing area in Languedoc, France, where the strain has showed the tolerance of high concen-tration of HMs (16–32 mM of Zn and 0.3–0.5 mM of Cd) at 28°C.”

Replace “are less as” with “is less as”.

“The adaptation of bacterial strain towards different stress such as cold temperature, high salt, and toxic HMs will be very valuable infor-mation in terms of understanding the bacteria from the polar region. Furthermore, these studies are helpful in exploring the diversity of HM-resistant microorganism, understand-ing the overlapping gap between biotic and abiotic processes, monitor environmental health and can be implemented for environmental and agricultural purposes in future.”

Add an “a” before bacterial strain and a “the” before future and replace “cold temperature” with “cold temperatures”, “microorganism” with “microorganisms” and “monitor” with “and monitoring”. 

Author Response

Authors’ Responses to the Review Comments

We are grateful for the insightful feedback and reviews provided by the editor and each reviewer. Reviewers’ remarks are shown in ‘black’, with detailed responses appearing in ‘blue’ and notes in ‘red’ color. For the revised main manuscript file, we have marked the modified text in yellow with reviewers' comments, and ‘green’ with English corrections.

Please note that the references are changed during revisions. In addition, sentences are corrected after English revision.

Reviewer 1

Comments and Suggestions for Authors

“Comparative genome analysis of polar Mesorhizobium sp. PAMC28654 to gain insight into tolerance for salinity and heavy metals stress”.

The manuscript contributes with a complete genome sequence of the strain Mesorhizobium PAMC28654 isolated from soils of Rwenzori Mountains in Uganda and proposes a genomic comparative study including 61 strains with available genome sequence in NCBI. Several annotation programs were used to accurately define the coding sequences related to environmental stresses of the PAMC28654. The genomic comparison was based on 16S rRNA phylogeny (Figure 3), statistics of the genome sequences (Tables 2 and 3), and genome annotation. In addition, the temperature, salt, and heavy metals tolerance, as well as nitrate reduction and EPS production of the strain PAMC28654 were evaluated. The main findings refer to the isolation of a Mesorhizobium strain from cold regions, an uncommon report in the literature, and the salt and heavy metals tolerance of this strain. The manuscript contains supplementary material; however, it is not available to download.

-Thank you for letting us know. The supplementary materials will be accessible for download.

Several tools and approaches may be applied to qualify a study as comparative genomics. Therefore, I suggest performing a robust core genome phylogeny based on phylogenomic analysis to demonstrate the relatedness of the strains; as well as better exploring the genome annotation of the closest strains and PAMC28654 in light of the features of interest trying to evidence the presence/absence of genes related to tolerance to stressful conditions.

-Thank you for your suggestion. We made additional attempts to identify our strain by using genome-based taxonomy. We compared our whole genome sequence with the whole genome sequences of all “type strain” of Mesorhizobium strains available using TYGS together with the MASH algorithm and (GBDP) approach. Furthermore, we also did ANI analysis using Orthologous Average Nucleotide Identity Tool (OAT).

Since the document with line numbers is not available, my comments are described here followed by the copied sentences from the manuscript, which were highlighted in the PDF document.

[Comment 1]:

Introduction section:

[Comment 1]: “The Mesorhizobium genus was identified by Jarvis et al., in 1997 [1] is a member of the family Phyllobacteriaceae, phylum Pseudomonadota, and class Alphaproteobacteria [2,3].”

Add an "and" before "is a member".

Thank you for the suggestion. We have inserted “and” before “is a member” as per suggestion.

[Comment 2]: “The main aim of the study was to isolate a new Mesorhizobium species from cold regions and study their possibilities for their survival in cold areas.”

The analysis provided in the manuscript did not allow to affirm that strain Mesorhizobium PAMC28654 is a new species, therefore the term “a new Mesorhizobium species” must be replaced with “a Mesorhizobium strain”.

-We appreciate the recommendation. "a Mesorhizobium strain" has been substituted for "a new Mesorhizobium species."

Please take note that a "Gram-negative bacterium" has been used in place of the term "new Gram-negative bacterium" in the abstract section.

[Comment 3]: “The strain Mesorhizobium sp. PAMC28654 was isolated from soil samples collected in the Rwenzori Mountains of Uganda by the Korea Polar Research Institute (KOPRI, Incheon, Korea).”

Please provide the geographic coordinates of the local where the strain was collected as well as more information about the isolation process (how the soil was collected, bacteria isolation method, culture media, growth conditions, pure colonies identification, etc).

-We appreciate you letting us know that we were overlooked. We have also included the strain's geographic coordinates of the local where the strain was collected, the method used to isolate the bacteria, the culture medium, the growth parameters, and the identification of pure colonies in the main manuscript.

[Comment 4]: “Similarly, we obtained 61 genomes of the Mesorhizobium species from the NCBI database.”

-  Several genomes from NCBI used in the study are from Mesorhizobium strains only classified to genus level, therefore the term “61 genomes of the Mesorhizobium species” must be replaced with “61 genomes of Mesorhizobium strains.” ;

-        Specify how the 61 genomic sequences were selected.

- Thank you for the suggestion. The phrase “61 genomes of the Mesorhizobium species” has been replaced with “61 genomes of Mesorhizobium strains”.

-Only the complete genome sequence was selected from the NCBI, so we rewrite the sentence “Similarly, we obtained 61 genomes of the Mesorhizobium strains from the NCBI database as “Similarly, we obtained 61 complete genomes of the Mesorhizobium strains from the NCBI database. 

[Comment 5]: “Phylogenetic tree was constructed using 16S rRNA sequences of the complete genomes of uncategorized and categorized Mesorhizobium sp. obtained from NCBI 16S rRNA were aligned using MUSCLE and MEGA 11 [22–27]”.

- Please replace “Mesorhizobium sp.” with “Mesorhizobium strains”;

- Provide the phylogenetic method and evolutionary model used to construct the tree and the number of re-samplings for bootstrap analysis.

-Thank you for the suggestion. "Mesorhizobium strains" has been used in place of "Mesorhizobium sp."

- We have used a neighbor-joining method and maximum composite likelihood model to construct a tree. Branch numbers represents percentages of bootstrap values in 1000 sampling replicates.

Please note: We rewrote the paragraph as “Phylogenetic trees were constructed using 16S rRNA sequences of complete genomes of uncategorized and categorized Mesorhizobium strains obtained from NCBI. These 16S rRNA sequences were aligned using MUSCLE, MEGA 11, and a neighbor-joining method [22–27]. Maximum composite likelihood model was used to construct the phylogenetic   tree. Branch numbers represents percentages of bootstrap values in 1000 sampling replicates. Related whole genome sequences of Mesorhizobium strains available in GenBank (https://www.ncbi.nlm.nih.gov) were also downloaded for identification and comparison with our strain. Genome-based taxonomic analysis was performed using a free bioinformatics platform Type (Strain) Genome Server (TYGS (https://tygs.dsmz.de) [28]. Based on TYGS results, similarity between strains was confirmed by comparing values of OrthoANI, which were calculated using the Orthologous Average Nucleotide Identity Tool (OAT) [29].

[Comment 6]: Table 2 and Table 3

- Since not all genomes from strains used in the manuscript are classified as a species, delete the “sp.” between “Mesorhizobium” and “strains” of the legend of both tables.

-Allocate Tables 2 and 3 in the supplementary material.

-Add the meaning of the N/A abbreviation in the footnotes of both tables.

-In the column “Host” of Table 2, change the species names to italic.

-Add a column with the accession number of genome sequences used in Table 2 of the manuscript.

-In the column “Protein count” of Table 3, put the total number of proteins.

-Indicate if the strain used is a type strain of a species by adding the superscript T (T) in the columns “Strain” and “Strain name” in Tables 2 and 3, respectively;

- Since not all genomes from strains used in the manuscript are classified as a species, replace “Mesorhizobium sp. strains” with “Mesorhizobium strains” from the legend of both tables.

-Thank you for the suggestion. We deleted the “sp.” between “Mesorhizobium” and “strains” of the legend of both tables;

-Table 2 and Table 3 were allocated in the supplementary materials.

-The footnotes of the two tables now include the meaning of the N/A abbreviation.

-A column with accession number of genome sequences was added in Table 2 of the manuscript.

-In the column “Protein count” of Table 3, was replaced by “total number of proteins”.

- The type strain in Tables 2 and 3, which will eventually become Supplementary Tables 1 and 2, now have the superscript T (T) appended in the columns for "Strain" and "Strain name."

- In the legend of both tables, "Mesorhizobium sp. strains" is substituted with "Mesorhizobium strains."

[Comment 7]: “Phylogenetic tree was constructed using 16S rRNA sequences of the complete ge-nomes of uncategorized and categorized Mesorhizobium sp. obtained from NCBI by using MEGA 11 (https://www.megasoftware.net/) showed that our strain is most closely re-lated to Mesorhizobium sp. INR15 (similarity of 99.87%, CP045496.1) […]”

Replace “Mesorhizobium sp.” with “Mesorhizobium strains”. The abbreviation sp. refers to a single species and spp. for several species. However, since not all genomes from strains used in the manuscript are classified as species, it must be properly fixed in the rest of the manuscript.

-We appreciate your suggestion. “Mesorhizobium sp.” has been replaced with “Mesorhizobium strains”. The remainder of the manuscript has the error corrected.

[Comment 8]: Phylogenetic comparison

Even though the 16S rRNA phylogeny is a technique broadly used in bacterial taxonomy, it provides information limited to genus level identification. It is hard to infer the relatedness of strains in a genus based uniquely on 16S rRNA phylogeny because the sequence of this gene is highly conserved. Since the manuscript proposes a comparative genomic study, and the authors selected a great number of complete genomic sequences of Mesorhizobium strains available in NCBI, I suggest constructing a robust phylogeny based on the core genome of the strains.

-Thank you for the suggestion. After the 16S rRNA phylogeny, we made additional attempts to identify our strain by using genome-based taxonomy. We compared whole genome sequence of our strain with the whole genome sequences of every “type strain” of all Mesorhizobium strains available using TYGS together with the MASH algorithm and (GBDP) approach. Furthermore, we also did ANI analysis using Orthologous Average Nucleotide Identity Tool (OAT). Data are shown in fig 3 (B) and fig 3 (C).

[Comment 9]: In the legend of Figure 3, add the size of the alignment of 16S rRNA sequences.

-Thank you for the suggestion. We added the size/length of the alignment of 16S rRNA sequences is 1496 nucleotides (approximately 1500 nucleotides).

Please note: We replaced the legend of Figure 3. with “Figure 3. (A) Neighbor-joining phylogenetic trees of 16S rRNA gene of Mesorhizobium sp. PAMC28654 with other Mesorhizobium strains without an outgroup using maximum composite likelihood model. The size of the alignment of 16S rRNA sequences is 1496 nucleotides (approximately 1500 nucleotides). Percentages in the bootstrap test are from 1,000 sample replicates. Only values above 50% are shown in branch nodes. (B) TYGS results for whole genome sequence of Mesorhizobium datasets. (C) Orthologous Average Nucleotide Identity (ANI) results of our strain Mesorhizobium sp. PAMC28654 with other selected Mesorhizobium strains available in the NCBI database.   

[Comment 10]: “Putative 3D structure modelling of the all three subunit Nar enzymes (NarG, α-subunit; NarH, β subunit; and NarI, γ subunit) were shown in (Supplementary Fig. S1)” and “The change in color to red from colorless solution was monitored. E. coli strain and our isolated strain showed change in color from colorless to red at both the temperatures 15°C and 25°C, confirming the reduction of nitrate whereas abiotic control did not show any color change (Supplementary Fig. S2).”

Supplementary material is not available to download.

-Thank you for letting us know. The supplementary materials will be accessible for download.

[Comment 11]: “Furthermore, other nitrogen met-abolic proteins (nrtP, nasA, narZ, and nxrA), nitrous-oxide reductase (nosZ) [EC:1.7.2.4], glutamine synthetase (glnA and glul) [EC:6.3.1.2], glutamate dehydrogenase (gudB and rocG) [EC:1.4.1.2], glutamate synthase (NADPH) large chain (gltB) [EC:1.4.1.13], carbamate kinase (arc) [EC:2.7.2.2], and carbonic anhydrase (CA) [EC:4.2.1.1], were identified.”

The protein abbreviation must be written with the first letter capitalized and not italic (e.g. NrtP), whereas the gene abbreviation must be written in lowercase and italic (e.g. nrtP - italic). Please fix the nomenclature of proteins/genes properly in the rest of the manuscript.

-Thank you for the indication and suggestion. We have reviewed the entire manuscript and corrected any errors related to protein and gene abbreviations. The gene abbreviation is written in lowercase and italics, while the protein abbreviation is written with the first letter capitalized and not italicized.

[Comment 12]: “Comparing the nitrogen metabolic enzymes of the complete genome of all the Mesorhizobium sp., nitrilase enzyme that convert the hydrolysis of nitriles to carboxylic acids and ammonia without the formation of free amide intermediates [45] were identified in Mesorhizobium sp. L-8-10, Mesorhizobium sp. L-8-3, Mesorhizobium sp.M1B.F.Ca.ET.045.04.1.1, Mesorhizobium sp. M1D.F.Ca.ET.043.01.1.1, two strains of Mesorhi-zobium sp. NZP2077, Mesorhizobium sp. Pch-S, and Mesorhizobium loti R88 but in remaining strain including our strain lack the nitrilase enzyme (Table 2)”.

It is not clear if all genomes retrieved from NCBI were annotated with the same programs as PAMC28654.

-Thank you for the indication. We annotated all of them using blastKOALA to create KEGG pathways and then compared.

Note: We rewrote the sentence as “When nitrogen metabolic enzymes of complete genomes of all Mesorhizobium strains were compared using blastKOALA, nitrilase enzyme that could convert nitriles to carboxylic acids and ammonia through hydrolysis without forming free amide intermediates [49]

Note: Table 2 has been replaced with supplementary Table 1 since it was assigned to the supplemental materials.

Note: References were also changed during corrections.

[Comment 13]: “Comparing the Mesorhizobium sps from Table 2. including draft genome of Mesorhizobium reported in the NCBI database so far HMs related work in the Mesorhizobium sps has been mostly done in copper, zinc, lead, cadmium, and chromium”.

The sentence is confusing, please rewrite it. Are the 61 genomic sequences draft or finished?

-Thank you for letting us know. We rewrote the sentence as “Comparing Mesorhizobium strains having complete genome from Supplementary Table 2. and also draft genomes of Mesorhizobium reported in the NCBI database and literatures so far HMs related work in the Mesorhizobium sps has been mostly done in copper, zinc, lead, cadmium, and chromium [14,28,52]”.

[Comment 14]: “In this study, we have isolated a new Mesorhizobium species, namely Mesorhizobium sp. PAMC28654 from the polar region of Uganda.”

Based on the results presented in the manuscript, it is not possible to affirm that PAMC28654 is a new species. 

-Thank you for the indication, suggestion, and valuable advice. We made additional attempts to identify our strain by using genome-based taxonomy. We compared whole genome sequence of our strain with the whole genome sequences of all “type strain” of Mesorhizobium strains available using TYGS together with the MASH algorithm and (GBDP) approach. Furthermore, we also did ANI analysis using Orthologous Average Nucleotide Identity Tool (OAT).

Please note, we added a paragraph in materials and methods “section 2.4. Phylogenetic comparison” “Maximum composite likelihood model was used to construct the phylogenetic tree. Branch numbers represents percentages of bootstrap values in 1000 sampling replicates. Related whole genome sequences of Mesorhizobium strains available in GenBank (https://www.ncbi.nlm.nih.gov) were also downloaded for identification and comparison with our strain. Genome-based taxonomic analysis was performed using a free bioinformatics platform Type (Strain) Genome Server (TYGS (https://tygs.dsmz.de) [28]. Based on TYGS results, similarity between strains was confirmed by comparing values of OrthoANI, which were calculated using the Orthologous Average Nucleotide Identity Tool (OAT) [29].’’

Besides that, we also added a paragraph in Results and Discussion section “3.2. Phylogenetic comparison, “We made additional attempts to identify our strain Mesorhizobium sp. PAMC28654 using genome-based taxonomy. We compared our whole genome sequence to whole genomes of all Mesorhizobium strains from a database of type strains using the MASH algorithm, a fast approximation of relatedness between genomes, the set of type-strains with the smallest MASH distance with our genome were observed/selected from TYGS [28]. Furthermore, a genome blast distance phylogeny (GBDP) approach was used to perform a pairwise comparison of strains that were closely related to our strain. The exact inter-genome distance could be inferred based on the trimming algorithm and distance. Results are shown in Figure 3B. We compared average nucleotide identity (ANI) values between a total of 19 strains, including our strain, the closest type lineages determined from the TYGS database, and 16S rRNA sequence results. These comparisons aimed to assess ANI values and determine bacterial species identification between our strain and the selected reference strains. Genome sequences’ ANI values ranged from 83.92% to 99.99% (Figure 3C). However, the ANI value obtained with complete genome of our strain was much lower than the typical ANI value of 96%. The ANI value of our strain was significantly lower than 96% when it was compared to other strains. Thus, our strain was not close to other strains. ANI analysis showed that the average nucleotide identity of all bacterial orthologous genes was shared between any two genomes. It offers a robust resolution between bacterial strains of the same or closely related species (i.e., species showing 80–100 % ANI) and identification between bacterial strains of the same or closely related species (i.e., species showing ANI values over 96%) [34, 35]. These results showed that our strain might belong to a new Mesorhizobium species. This strain was deposited in the NCBI database as Mesorhizobium sp. PAMC28654.’’

Furthermore, we included additional data in Fig 3 (B) and 3 (C).

Comments on the Quality of English Language

The manuscript presents some grammar mistakes, and it should be carefully reviewed. Please review the English in the following sentences: 

We appreciate you pointing out some grammatical errors in our manuscript. We corrected those grammar mistakes as suggested by the reviewer 1. In addition, we also send our manuscript for English correction. We recorrected sentences with grammatical errors and spellings. The corrections are highlighted in green color in the main manuscript.

“The broad distribution of this genus and its ability to make a symbiotic relationship to several plant genera makes it an interesting candidate for agronomic and ecological pro-spective [7,8].”

Replace “to” with “with”.

-Thank you for the suggestion. We replace “to” with “with” in the sentence “The broad distribution of this genus and its ability to make a symbiotic relationship to several plant genera makes it an interesting candidate for agronomic and ecological prospective [7,8].”

“In addition to that, use of EPS produced by rhizobium such as Mesorhizobium has contributed significantly to industrial purposes as their use as a gelling, thickening, and stabilizing agents in foods, pharmaceutical, and cosmetics [9].”

-        Add a “the” before use.

-        Replace “pharmaceutical” with “pharmaceuticals”.

 -We appreciate your suggestion. We added “the” before use.

-We replaced “pharmaceutical” with “pharmaceuticals” in the sentence “In addition to that, use of EPS produced by rhizobium such as Mesorhizobium has contributed significantly to industrial purposes as their use as a gelling, thickening, and stabilizing agents in foods, pharmaceutical, and cosmetics [9].”

“Furthermore, abiotic stresses such as heavy metals (HMs), temperature stress, salinity, nutrient availability stress has been reported in the polar areas. “

-Add an “and” between salinity and nutrient availability.

-Replace “has” with “have”.

-Thank you for the suggestion. We added “and” between salinity and nutrient availability.

-We replaced “has” with “have”.

“Bacteria has been reported to tolerate a wide range of temperatures from extreme high temperature (thermophiles) to extreme low temperature (psychrophiles). Bacteria mediate the ability to adjust to low or cold temperature by structural adjustment of enzymes, maintenance of membrane fluidity, expression of cold shock protein, and adaptation of the translation and transcription machinery [10,12]. “

-        Replace “has” with “have”.

-        Replace the two “extreme” with “extremely”.

-        Replace “temperature” with “temperatures”.

 -Thank you for the suggestion. We replaced “has” with “have”.

-We replaced “extreme” with “extremely”.

-We replaced “temperature” with “temperatures”.

“Next to temperature, HMs (those elements having high atomic weight and atomic number such as copper, Cu; zinc, Zn; cadmium, Cd; arsenate, As; lead, Pb; mercury, Hg; chromium, Cr; and nickel, Ni) stress is another factor that has huge influence on bacterial survival.”

Replace “has huge” with “have a huge”.

 -Thank you for the suggestion. We replaced “has huge” with “have a huge.”

“Besides temperature and HMs stress, salt stress is yet another factor that have huge impact on the bacterial growth and survival, and biggest hurdle in achieving better crop yield and quality with soil having high salinity [16]. “

- Replace “have huge” with “has a huge”.

- Add a “the” before biggest.

-Thank you for the suggestion. We replaced “have huge” with “has a huge”.

- We added “the” before biggest.

“Even though, Mesorhizobium species has been studied mostly at the optimum temperature (25°C–30°C) [17] for agronomic, ecological, and industrial applica-tion [18], and very limited studies has been carried out for extreme environmental condition as low temperature particularly for issues like HMs and salinity. “

Replace “has been studied” with “have been studied” and “has been carried” with “have been carried”.

 -Thank you for the suggestion. We replaced “has been studied” with “have been studied” and “has been carried” with “have been carried”.

“This study is very helpful to a better understanding of bacteria mediating their survival on cold temperature as well as other stress, such as salinity and HMs toxicity through the production of EPS. In addition to that, the isolated strain might be implemented in the future for environmental and agricultural aspects.”

Replace “cold temperature” with “cold temperatures”.

-Thank you for the suggestion. We replaced “cold temperature” with “cold temperatures”.

The isolated strains were checked for their growth in several media, including R2A, lysogeny agar (LBA), nutrient agar (NA), tryptic soy agar (TSA), and marine agar (MA) at three different temperature 15°C, 25°C, and 37°C. “

Replace “different temperature” with “different temperatures”.

 -Thank you for the suggestion. We replaced “different temperature with “different temperatures” in the manuscript wherever we found error.

“The putative 3D structure of nitrate reductase was modeled by online program PHYRE2 Server (http://genome3d.eu/) in the intensive mode based on the most identical template in the PDB database [28].”

Add a “the” before online program.

-Thank you for the suggestion. We added “the” before the online program in the sentence “The putative 3D structure of nitrate reductase was modeled by online program PHYRE2 Server (http://genome3d.eu/) in the intensive mode based on the most identical template in the PDB database.

“For the confirmation, a pinch of zinc dust was added to the tube with reagent A and B. “

Replace “reagent” with “reagents”.

 -Thank you for the suggestion. We replaced “reagent” have been replaced with “reagents”. “For the confirmation, a pinch of zinc dust was added to the tube with reagent A and B.’’

“The isolated strain was tested for EPS production at both temperature (15°C and 25°C). “

Replace “temperature” with “temperatures” in the sentence, as well as in the rest of the manuscript when it is describing two or more temperatures.

  -Thank you for the suggestion. “temperature” has been replaced with “temperatures” in the sentence, as well as in the rest of the manuscript.

“Negative control without the bacteria and positive control using the bacteria were also used. The bacterial growth OD600 was measured by spectrophotometer. Initial concentration of bacterial cells used for the salinity toxicity was ˜2.42 × 108 cells/ml (reference E. coli cells, OD600 of 1.0 = 8 × 108 cells/ml). “

Add an “a” before spectrophotometer and a “The” before initial.

-Thank you for the suggestion. “a” has been added before spectrophotometer and a “The” before initial. 

“Initial concentration of bacterial cells used for the HMs toxicity was ˜2.42 × 108 cells/ml (reference E. coli cells, OD600 of 1.0 = 8 × 108 cells/ml).”

Add a “The” before initial.

-Thank you for the suggestion. “The” has been added before initial in “Initial con-centration of bacterial cells used for the HMs toxicity was ˜2.42 × 108 cells/ml (reference E. coli cells, OD600 of 1.0 = 8 × 108 cells/ml).”

“The isolated strains Mesorhizobium sp. PAMC28654 showed its growth in R2A broth and nutrient agar (NA) at two different temperature 15°C and 25°C, besides that, the strain showed their growth at specific media such as in the nitrate broth and the nitrite broth at 15°C and 25°C. “

Replace “strains” with “strain” and “temperature” with “temperatures”.

-Thank you for the suggestion. “strains” has been replaced with “strain”.

“Nar enzymes have been reported to be responsible for the conversion of initial step in the nitrate reduction pathway and often in anaerobic or low-oxygen condi-tion. “

Add a “the” before initial.

-Thank you for the suggestion. “strains” has been replaced with “strain”.

“NarG is responsible for binding and reducing nitrate to nitrite, which facilitates the reduction of nitrate by transferring the electrons to NarG, and NarI, anchor the NarG and NarH subunit to the bacterial membrane.”

Replace “anchor” with “anchoring”.

-Thank you for the suggestion. ”anchor has been replaced with “anchoring” in the sentence in the manuscript “NarG is responsible for binding and reducing nitrate to nitrite, which facilitates the reduction of nitrate by transferring the electrons to NarG, and NarI, anchor the NarG and NarH subunit to the bacterial membrane.”

“ In addition to that, nitrate/nitrite protein (Nark)responsible for importing nitrate from the extracellular environment into the bacterial cell to be utilized by the nitrate reductase complex.”

Remove the comma after (NarK) and add “is” before responsible.

-Thank you for the suggestion. A comma was omitted following (NarK), and "is" was added.

“Putative 3D structure modelling of the all three subunit Nar enzymes (NarG, α-subunit; NarH, β subunit; and NarI, γ subunit) were shown in (Supplementary Fig. S1). “

Remove “all” before three.

 -That you for the suggestion. We removed “all” before “three” in the sentence “Putative 3D structure modelling of the all three subunit Nar enzymes (NarG, α-subunit; NarH, β subunit; and NarI, γ subunit) were shown in (Supplementary Fig. S1). “

“Even though, our strain showed some nitrogen metabolic enzymes such as Nar but lack of further denitrification proteins which coincide with many microorganisms that has been well reported to reduce nitrate to nitrite but lack further denitrification enzymes [43,44]. “

-        Replace “lack of” with “lacked”.

-        Replace “has been well reported” with “have been well reported”.

- Thank you for the suggestion. We replaced “Lack of” with “lacked”.

- We replaced “has been well reported” with “have been well reported”.

“[…] but in remaining strain including our strain lack the nitrilase enzyme (Table 2).”

Replace “strain” with “strains”.

 -Thank you for the suggestion. We replaced “strain” with strains”.

“The change in color to red from colorless solution was monitored. E. coli strain and our isolated strain showed change in color from colorless to red at both the temperatures 15°C and 25°C, confirming the reduc-tion of nitrate whereas abiotic control did not show any color change (Supplementary Fig. S2).”

Add an “a” before change in color.

-Thank you for the suggestion. We added “a” before change in color” in the sentence “The change in color to red from colorless solution was monitored. E. coli strain and our isolated strain showed change in color from colorless to red at both the temperatures 15°C and 25°C, confirming the reduc-tion of nitrate whereas abiotic control did not show any color change (Supplementary Fig. S2).”

The genome analysis of the isolated strain showed EPS-producing protein ExoF and ExoQ. “

Replace “protein” with “proteins”.

 -Thank you for the suggestion. We replaced “protein” with “proteins”.

“The weight of EPS was little bit higher at 15°C than at 25°C, which complies with the study done by Perviaz ali et al., [46] on Pseu-domonas sp. “

Add a “a” before little.

-Thank you for the suggestion. “a” has been added to the “The weight of EPS was little bit higher at 15°C than at 25°C, which complies with the study done by Perviaz ali et al., [46] on Pseu-domonas sp. “

“The strain showed tolerance of NaCl up to 100 mM at both the temperature at 15°C to 25°C. However, no difference in salt tolerance concentration was observed at both temperatures. “

Replace “both the temperature” with “both temperatures”.

-Thank you for the suggestion. We replaced “both the temperature” with “both temperatures” in the sentence “The strain showed tolerance of NaCl up to 100 mM at both the temperature at 15°C to 25°C. However, no difference in salt tolerance concentration was observed at both tem-peratures. “

“In addition to that, some Mesorhizobium sps isolated from acacias in arid and semi-arid regions in Al-geria has shown to variability in the tolerance of NaCl. “

Replace “has” with “have” and remove “to” before variability.

-Thank you for the suggestion. We replaced has with have in the sentence “In addition to that, some Mesorhizobium sps isolated from acacias in arid and semi-arid regions in Al-geria has shown to variability in the tolerance of NaCl.

“Salt tolerance and dependence are the characteristics of some micro-organism and microorganism-having potential to tolerate salt stress therefore could be implemented for agricultural purpose to overcome detrimental effect and management of saline soil for better crop productivity [16,49] considering that, even though our bacterial strain did not show differences in salt tolerant at two different temperatures, but still the strain has the capacity to tolerate NaCl up to 100 mM so the strain might have the potential to implemented for agricultural purpose in future needs further study.”

- Put “therefore” between commas, replace “purpose” with “purposes”, “effect” with “effects” and “tolerant” with “tolerance”.

-Add a “to” between “to” and “implemented” and replace “purpose” with “purposes”.

-Thank you for the suggestion. We added “therefore” between commas, and also we replaced “purpose” with “purposes”, “effect” with effects” and “tolerant” with “tolerance” in the sentence “Salt tolerance and dependence are the characteristics of some micro-organism and microorganism-having potential to tolerate salt stress therefore could be implemented for agricultural purpose to overcome detrimental effect and management of saline soil for better crop productivity [16,49] considering that, even though our bacterial strain did not show differences in salt tolerant at two different temperatures, but still the strain has the capacity to tolerate NaCl up to 100 mM so the strain might have the potential to implemented for agricultural purpose in future needs further study.”

“Microbes confer various types of resistance mechanism in response to HMs [50] and efflux pumps are the major known players for group of resistance systems with both plasmid and chromosomal system [51]. “

Put “mechanism” and “system” in the plural.

-Thank you for the suggestion. We put “mechanism” and system in plural “mechanisms” and system as systems.

“HM resistant protein such as copper resistant protein CopC, CopD, arsenical resistant protein ACR3, cobalt, zinc, and cadmium resistant pro-tein were identified from the genome analysis (Table 4) of the isolated strain so, only those HMs were considered and subjected to the wet-lab experiments. “

Replace “HM resistant protein” with “HM resistant proteins”.

-Thank you for the indication. We replaced “HM resistant protein” with “HM resistant proteins”.

“The result demonstrated difference in different HMs tolerance ability of our isolated strain at two different temperatures (15°C and 25°C). “

Replace “difference” with “differences”.

-Thank you for the suggestion. We replaced “difference” with “differences.”

“The resistance capacity of our strain for HMs such as zinc and cad-mium are less as compared to the study done by Celine Vidal et al., [14] on Mesorhizobium metallidurans belonging to the same genus isolated from the metallicolous soil of zinc min-ing area in Languedoc, France, where the strain has showed the tolerance of high concen-tration of HMs (16–32 mM of Zn and 0.3–0.5 mM of Cd) at 28°C.”

Replace “are less as” with “is less as”.

-Thank you for the suggestion. We replaced “are less as” with “is less as” in the sentence “The resistance capacity of our strain for HMs such as zinc and cad-mium are less as compared to the study done by Celine Vidal et al., [14] on Mesorhizobium metallidurans belonging to the same genus isolated from the metallicolous soil of zinc min-ing area in Languedoc, France, where the strain has showed the tolerance of high concen-tration of HMs (16–32 mM of Zn and 0.3–0.5 mM of Cd) at 28°C.”

“The adaptation of bacterial strain towards different stress such as cold temperature, high salt, and toxic HMs will be very valuable information in terms of understanding the bacteria from the polar region. Furthermore, these studies are helpful in exploring the diversity of HM-resistant microorganism, understanding the overlapping gap between biotic and abiotic processes, monitor environmental health and can be implemented for environmental and agricultural purposes in future.”

Add an “a” before bacterial strain and a “the” before future and replace “cold temperature” with “cold temperatures”, “microorganism” with “microorganisms” and “monitor” with “and monitoring”. 

-Thank you for the suggestion. We added “a” before bacterial strain and a “the” before future. We replaced “cold temperature” with “cold temperatures”, “microorganism” with “microorganisms” and “monitor” with “and monitoring”. 

Reviewer 2 Report

Comments and Suggestions for Authors

The study of Khanal and co-authors "Comparative genome analysis of polar Mesorhizobium sp. PAMC28654 to gain insight into tolerance for salinity and heavy metals stress"  is relevant, the experimental scheme is correct, and the methods are suitable and modern. The obtained results have scientific significance, and the article will be of interest to readers.

However, some minor revision is required, including text editing and improving the quality of the figures' presentation (see my comments in the attached PDF file).

Best wishes 

Author Response

Authors’ Responses to the Review Comments

We are grateful for the insightful feedback and reviews provided by the editor and each reviewer. Reviewers’ remarks are shown in ‘black’, with detailed responses appearing in ‘blue’ and note in ‘red’ color. For the revised main manuscript file, we have marked the modified text in yellow with reviewers' comments, and ‘green’ with English corrections.

Please note that the references are changed during revisions. In addition, sentences are corrected after English revision.

Reviewer 2

Comments and Suggestions for Authors

The study of Khanal and co-authors "Comparative genome analysis of polar Mesorhizobium sp. PAMC28654 to gain insight into tolerance for salinity and heavy metals stress" is relevant, the experimental scheme is correct, and the methods are suitable and modern. The obtained results have scientific significance, and the article will be of interest to readers.

However, some minor revision is required, including text editing, and improving the quality of the figures' presentation (see my comments in the attached PDF file).

Best wishes 

Give the full name of the first mention of “EPS”.

In addition to that, use of EPS produced by rhizobium such as Mesorhizo-bium has contributed significantly to industrial purposes as their use as a gelling, thicken-ing, and stabilizing agents in foods, pharmaceutical, and cosmetics [9].

-Thank you so much for the suggestion. We substitute EPS with the full name Exopolysachharide in the sentence “In addition to that, use of EPS produced by rhizobium such as Mesorhizo-bium has contributed significantly to industrial purposes as their use as a gelling, thicken-ing, and stabilizing agents in foods, pharmaceutical, and cosmetics [9].

Give the full name by the first mention of “HMs”.

The EPS–producing bacteria might have potential for adaptive strategy by management of multiple abiotic stress such as temperature stress, HMs stress, and salinity stress [10].

Polar regions, being an isolated environment and having an extremely harsh climate have been reported to be affected by increased human activities including climate change and global warming. Furthermore, abiotic stresses such as heavy metals (HMs), tempera-ture stress, salinity, nutrient availability stress has been reported in the polar areas. The microorganisms living in these areas have been reported to have developed adaptive strategies to survive in extreme conditions and resist varieties of abiotic stresses [11].

-We write the full name by the first mention of “HMs” as “heavy metals (HMs)”. We replaced “heavy metals (HMs)” with “HMs” as per suggestion.

Check the font.

Change the The annotation of the genome was per-formed using the NCBI Prokaryotic Genome Annotation Pipeline (PGAP). Detailed infor-mation about PGAP can be found at https://www.ncbi.nlm.nih.gov/genome/annota-tion_prok/).

-Thank you for the suggestion. We changed the font according to the journal format in https://www.ncbi.nlm.nih.gov/genome/annota-tion_prok/).

The putative 3D structure of nitrate reductase was modeled by online program PHYRE2 Server (http://genome3d.eu/) in the intensive mode based on the most identical template in the PDB database [28].

-Thank you for the suggestion. We changed the font according to the journal format in (http://genome3d.eu/)

Improve the quality of graph in figure 6. Extra space after the graph in fig 5.

-Thank you for the suggestion. We replaced the better quality of the graph in figures 6 and also removed extra space after the graph in fig 5.

Note: We replaced the figures and graph with better quality and did text editing as per your suggestion.

Round 2

Reviewer 1 Report

Comments and Suggestions for Authors

Comments on the Quality of English Language

Author Response

Authors’ Responses to the Review Comments

We are grateful for the insightful feedback and 2nd reviews provided by the reviewer. Reviewers’ remarks are shown in ‘black’, with detailed responses appearing in ‘blue’ and notes in ‘red’ color. For the revised main manuscript file, we have marked the modified text in Pink color after 2nd revision.

Reviewer 1

Comments and Suggestions for Authors

“Comparative genome analysis of polar Mesorhizobium sp. PAMC28654 to gain insight into tolerance for salinity and heavy metals stress”

The study describes the complete genome of Mesorhizobium PAMC28654 and several insights into tolerance to abiotic stresses and nitrogen metabolism. The authors followed all the suggestions recommended in the first review report. I still have some comments that I described below.

Also, according to the information for authors of the Microorganisms journal, the abstract should have 200 words at maximum and figures should be provided in a single zip archive with adequate resolution (1000 pixels or 300 dpi or higher). Since its abstract has 224 words and the figures are allocated in the manuscript, please, adjust it accordingly.

-Thank you for letting us know. The abstract now has a word count of under 200 (exact word count = 184). We sent figures with a sufficient resolution of over 300 dpi in a single zip archive. It currently has 600 dpi.

Comment 1) Please remove the sentence “a Gram-negative bacterium belonging to genus Mesorhizobium”, it may be redundant.

-Thank you for the suggestion. We removed the sentence “a Gram-negative bacterium belonging to genus Mesorhizobium” from the main manuscript.

Comment 2) Remove “habitat” and replace “environment” with “environments” in the sentence: “They have been isolated worldwide from terrestrial habitat environment especially from soil and root nodules”.

-Thank you for the suggestion. We removed “habitat” and replaced it with “environments”.

Comment 3) Replace “from” with “for” in the sentence “The broad distribution of this genus and its ability to make a symbiotic relationship with several plant genera make it an interesting candidate from agronomic and ecological prospectives [7,8].”

-Thank you for the suggestion. We replaced “from” with “for” in the sentence “The broad distribution of this genus and its ability to make a symbiotic relationship with several plant genera make it an interesting candidate from agronomic and ecological prospectives [7,8]”.

Comment 4) Replace “application” with “applications” in the sentence “Although Mesorhizobium species have been studied mostly at optimum temperatures (25°C–30°C) [17] for agronomic, ecological, and industrial application [18] […]”.

-Thank you for the suggestion. We replaced “application” with “applications” in the sentence “Although Mesorhizobium species have been studied mostly at optimum temperatures (25°C–30°C) [17] for agronomic, ecological, and industrial application [18] […]”.

Comment 5) Replace “of isolated strain” with “of the isolated strain” in the sentence “This study also aimed to determine specific genes and some special features of isolated strain in terms of their tolerance to HM and salinity.”

-Thank you for the suggestion. We replaced “of isolated strain” with “of the isolated strain” in the sentence “This study also aimed to determine specific genes and some special features of isolated strain in terms of their tolerance to HM and salinity.”

Comment 6) Allocate the following sentences “Annotation of the genome was performed using NCBI Prokaryotic Genome Annotation Pipeline (PGAP). Detailed information about PGAP can be found at https://www.ncbi.nlm.nih.gov/genome/annotation_prok/. Furthermore, coding DNA sequences (CDSs) were predicted and annotated with the Rapid Annotation using Subsys- tem Technology (RAST) server [20] available at https://rast.nmpdr.org/. Complete genome of the Mesorhizobium sp. PAMC28654 was visualized using the CGView server in Proksee (https://proksee.ca/) to generate a circular map [21].” from topic 2.2 into topic 2.3.

-Thank you for the suggestion. We allocated the following sentences. “Annotation of the genome was performed using NCBI Prokaryotic Genome Annotation Pipeline (PGAP). Detailed information about PGAP can be found at https://www.ncbi.nlm.nih.gov/genome/annotation_prok/. Furthermore, coding DNA sequences (CDSs) were predicted and annotated with the Rapid Annotation using Subsys- tem Technology (RAST) server [20] available at https://rast.nmpdr.org/. Complete genome of the Mesorhizobium sp. PAMC28654 was visualized using the CGView server in Proksee (https://proksee.ca/) to generate a circular map [21].” from topic 2.2 into topic 2.3.

Please note: We rewrote the whole paragraph in the manuscript as “In our study, we employed a diverse set of annotation tools to investigate genomic characteristics of strain PAMC28654 comprehensively. Annotation of the genome was performed using NCBI Prokaryotic Genome Annotation Pipeline (PGAP). Detailed information about PGAP can be found at https://www.ncbi.nlm.nih.gov/genome/annotation_prok/.Initially, the genome of PAMC28654 was annotated with Rapid Annotation using Subsystem Technology (RAST) server [20] available at https://rast.nmpdr.org/. Furthermore, were predicted. The RAST server, enabling the identification and annotation of genes along with their associated functions as well as coding DNA sequences (CDSs). Subsequently, predicted gene sequences were translated and subjected to an extensive search across multiple databases, including the National Center for Biotechnology Information (NCBI) non-redundant database, UniProtKB/Swiss-Prot, and Protein Data Bank proteins (PDB). This comprehensive annotation approach allowed us to gather a thorough understanding of genomic features of strain PAMC28654. By leveraging multiple databases and annotation tools, we could validate and enhance annotations, ensuring reliability and accuracy of identified genes and their potential functions. By employing this meticulous annotation strategy, we unveiled crucial insights into the genetic makeup of strain PAMC28654 and gained a deeper understanding of its biological potential and function attributes. Complete genome of the Mesorhizobium sp. PAMC28654 was visualized using the CGView server in Proksee (https://proksee.ca/) to generate a circular map [21].

Comment 7) Please replace “underwent annotation using” with “was annotated with” in the sentence “Initially, the genome of PAMC28654 underwent annotation using the RAST server […]”.

-We appreciate your suggestion. We replaced “underwent annotation using” with “was annotated with” in the sentence “Initially, the genome of PAMC28654 underwent annotation using the RAST server […]”.

Comment 8) Replace “Isolated strain” with “The isolated strain” in the sentence “Isolated strain was checked for their growth in several media, including R2A, lysogeny agar (LBA), nutrient agar (NA), tryptic soy agar (TSA), and marine agar (MA), at three different temperatures of 15°C, 25°C, and 37°C”.

-Thank you for the suggestion. We replaced “Isolated strain” with “The isolated strain” in the sentence “Isolated strain was checked for their growth in several media, including R2A, lysogeny agar (LBA), nutrient agar (NA), tryptic soy agar (TSA), and marine agar (MA), at three different temperatures of 15°C, 25°C, and 37°C”.

Comment 9) Remove the second “at” in the sentence “Strain Mesorhizobium sp. PAMC28654 was cultured in a nitrate broth at three different temperatures at 15°C, 25°C, and 37°C for 2 days”.

-Thank you for letting us know. We removed the second “at” in the sentence “Strain Mesorhizobium sp. PAMC28654 was cultured in a nitrate broth at three different temperatures at 15°C, 25°C, and 37°C for 2 days”.

Comment 10) Replace “E. Coli” with “Escherichia coli” in this sentence. Also, add a point after “included”in the sentence “An abiotic control without any microorganism and a positive control with E. Coli strain were also included”.

-We appreciate your suggestion. We replaced “E. Coli” with “Escherichia coli”. In addition, a point was “included” in the sentence “An abiotic control without any microorganism and a positive control with E. Coli strain were also included”.

Comment 11) Replace “the bacteria” with “the bacterium” in the sentence “To test for EPS production by the bacteria, [...]”.

-Thank you for letting us know. We replaced “the bacteria” with “the bacterium” in the sentence “To test for EPS production by the bacteria, [...]”.

Comment 12) Replace “describe” with “described” in the sentence “EPS was extracted as describe previously [31]”.

-Thank you for letting us know. We replaced “describe” with “described” in the sentence “EPS was extracted as describe previously [31]”.

Comment 13) Should the positive control be a strain that is known as salt tolerant instead of the Mesorhizobium PAMC28654, which is the strain that you are testing?

-Thank you for the indication, suggestion, and valuable advice. In the case of the nitrate reduction experiment, we used positive control as E. Coli., as they are well known to be responsible for nitrate reduction and are easily available. The nitrate reduction test was only a qualitative test.

The salinity test is a quantitative test. In case of salinity experiment, (1) control (Mesorhizobium sp. PAMC28654 bacterial cells were added in a culture tube containing media but without NaCl) and (2) test (Mesorhizobium sp. PAMC28654 bacterial cells were added in a culture tube containing media with NaCl) and were incubated at two different temperatures of 15°C and 25°C for 7 days. We corrected the error using the wrong choice of word during the writing of the manuscript. We replaced the word “positive control” with the word “control” in the case of salinity experiment. We added the sentence “Furthermore, for the salinity experiment, (1) control (Mesorhizobium sp. PAMC28654 bacterial cells were added in a culture tube containing media but without NaCl) and (2) test (Mesorhizobium sp. PAMC28654 bacterial cells were added in a culture tube containing media with NaCl) and were incubated at two different temperatures of 15°C and 25°C for 7 days.’ in the materials and methods section 2.8. in the manuscript for clear understanding.

Please note: In the process of revising the manuscript, we found the same type of error of using a wrong choice of word in case of HMs experiments (quantitative test), so we replaced the words “positive control with “control”.

In addition, we also added the sentence “Furthermore, for the HMs experiment, (1) control (Mesorhizobium sp. PAMC28654 bacterial cells were added in a culture tube containing media but without HMs) and (2) test (Mesorhizobium sp. PAMC28654 bacterial cells were added in a culture tube containing media with HMs) and were incubated at two different temperatures of 15°C and 25°C for 7 days” in the materials and methods section 2.9. Additionally, we changed the words in fig 5 and fig 6 by replacing the words “positive control” with “control” and what does that control mean. The error was only during the manuscript writing. Only the word has been changed not the data and graph. We also added the sentence in the legend of fig 5 and fig 6 for more clarity.

Comment 14) Replace “The circular map and total genome information of Mesorhizobium sp. PAMC28654 are shown in Figure 1” with “The genome information of Mesorhizobium sp. PAMC28654 and the circular map are shown in Table 1 and Figure 1, respectively”.

-Thank you for the suggestion. “The circular map and total genome information of Mesorhizobium sp. PAMC28654 are shown in Figure 1” with “The genome information of Mesorhizobium sp. PAMC28654 and the circular map are shown in Table 1 and Figure 1, respectively”.

Comment 15) Remove the row referring to the number of scaffolds in Table 1.

-Thank you for the suggestion. We removed the row referring to the number of scaffolds in Table 1.

Comment 16) What do you mean by categorized and uncategorized strains in the sentence “Among these genomes, 39 strains remained uncategorized, while 22 strains had been categorized based on available information”?

-Thank you for letting us know the errors in the sentence. We removed the sentence from the main manuscript in the paragraph.

Comment 17) Remove the sentence “Table 2 and Table 3 provide detailed information regarding the Mesorhizobium genus, facilitating a comprehensive overview of available genome information”.

Comment 18) Replace “Mesorhizobium sp.” with “Mesorhizobium strains” and remove

“(https://www.megasoftware.net/)” in the sentence “A phylogenetic tree was constructed using 16S rRNA sequences of complete genomes of uncategorized and categorized Mesorhizobium sp. obtained from NCBI using MEGA 11 (https://www.megasoftware.net/)”.

Comment 19) Replace “identify” with “evaluate” in the sentence “We made additional attempts to identify our strain Mesorhizobium sp. PAMC28654 using genome-based taxonomy”.

Comment 20) Remove the hyphen between “type” and “strain” and replace “were” with “was” in the sentence “We compared our whole genome sequence to whole genomes of all Mesorhizobium strains from a database of type strains using the MASH algorithm, a fast approximation of relatedness between genomes, the set of type-strains with the smallest MASH distance with our genome were observed/selected from TYGS [28]”.

-Thank you so much for the suggestion. We removed the hyphen between “type” and “strain”. We also replaced “were” with “was” in the sentence “We compared our whole genome sequence to whole genomes of all Mesorhizobium strains from a database of type strains using the MASH algorithm, a fast approximation of relatedness between genomes, the set of type-strains with the smallest MASH distance with our genome were observed/selected from TYGS [28]”.

Comment 21) Add a “the” before “remaining” in the sentence “However, remaining strains including our strain lacked the nitrilase enzyme (Supplementary Table 1)”.

-Thank you so much for the suggestion. We added a “the” before “remaining” in the sentence “However, remaining strains including our strain lacked the nitrilase enzyme (Supplementary Table 1)”.

Comment 22) Replace “work” with “studies” in the sentence “Most of the work have been done in terms of nitrogen fixation and nodulation”.

-Thank you so much for the suggestion. We replaced “work” with “studies” in the sentence “Most of the work have been done in terms of nitrogen fixation and nodulation”.

Comment 23) Describe the bacteria used in the positive control of the salinity test.

-Thank you for the indication, suggestion, and valuable advice. In the case of the nitrate reduction experiment, we used positive control as E. Coli., as they are well known to be responsible for nitrate reduction and are easily available. The nitrate reduction test was only a qualitative test.

The salinity test is a quantitative test. In case of salinity experiment, (1) control (Mesorhizobium sp. PAMC28654 bacterial cells were added in a culture tube containing media but without NaCl) and (2) test (Mesorhizobium sp. PAMC28654 bacterial cells were added in a culture tube containing media with NaCl) and were incubated at two different temperatures of 15°C and 25°C for 7 days. We corrected the error using the wrong choice of word during the writing of the manuscript. We replaced the word “positive control” with the word “control” in the case of salinity experiment. We added the sentence “Furthermore, for the salinity experiment, (1) control (Mesorhizobium sp. PAMC28654 bacterial cells were added in a culture tube containing media but without NaCl) and (2) test (Mesorhizobium sp. PAMC28654 bacterial cells were added in a culture tube containing media with NaCl) and were incubated at two different temperatures of 15°C and 25°C for 7 days.’ in the materials and methods section 2.8. in the manuscript for clear understanding.

Please note: In the process of revising the manuscript, we found the same type of error of using a wrong choice of word in case of HMs experiments (quantitative test), so we replaced the words “positive control with “control”.

In addition, we also added the sentence “Furthermore, for the HMs experiment, (1) control (Mesorhizobium sp. PAMC28654 bacterial cells were added in a culture tube containing media but without HMs) and (2) test (Mesorhizobium sp. PAMC28654 bacterial cells were added in a culture tube containing media with HMs) and were incubated at two different temperatures of 15°C and 25°C for 7 days” in the materials and methods section 2.9. Additionally, we changed the words in fig 5 and fig 6 by replacing the words “positive control” with “control”. Only the word has been changed not the data and graph. We also added the sentence in the legends of fig 5 and fig 6 for more clarity.

Comment 24) Replace “Mesorhizobium sp.” with “Mesorhizobium strains” and remove the comma before “are compared” in the sentence “When Mesorhizobium sp. (Supplementary Table 1), are compared, in the sentence “When Mesorhizobium sp. (Supplementary Table 1), are compared, Mesorhizobium loti MAFF303099 has been reported to be salt-sensitive (28% growth with 1.71 mM of NaCl) [51]”.

-Thank you for the suggestion. We replaced “Mesorhizobium sp.” with “Mesorhizobium strains” and we also removed the comma before “are compared” in the sentence “When Mesorhizobium sp. (Supplementary Table 1), are compared, Mesorhizobium loti MAFF303099 has been reported to be salt-sensitive (28% growth with 1.71 mM of NaCl) [51]”.

Comment 25) Replace “bacteria” with “strain” in the sentence “[...] it was found that Mesorhizobium sp. PAMC28654 bacteria could tolerate 1 mM CuSO4·5H2O, 2 mM CoCl2·6H2O, 1 mM ZnSO4·7H2O, 0.05 mM, Cd(NO3)2·4H2O, and 100 mM Na2HAsO4·7H2O at 15°C and 0.25 mM CuSO4·5H2O, 2 mM CoCl2·6H2O, 0.5 mM ZnSO4·7H2O, 0.01 mM Cd(NO3)2·4H2O, and 100 mM Na2HAsO4·7H2O at 25°C (Figure 6)”.

-Thank you for the suggestion. We replaced ““bacteria” with “strain” in the sentence “[...] it was found that Mesorhizobium sp. PAMC28654 bacteria could tolerate 1 mM CuSO4·5H2O, 2 mM CoCl2·6H2O, 1 mM ZnSO4·7H2O, 0.05 mM, Cd(NO3)2·4H2O, and 100 mM Na2HAsO4·7H2O at 15°C and 0.25 mM CuSO4·5H2O, 2 mM CoCl2·6H2O, 0.5 mM ZnSO4·7H2O, 0.01 mM Cd(NO3)2·4H2O, and 100 mM Na2HAsO4·7H2O at 25°C (Figure 6)”.

Comment 26) Add an “a” before “zinc mining” in the sentence “[...] isolated from a metallicolous soil of zinc mining area in Languedoc, France reported by Vidal et al. [14]”.

-Thank you for the suggestion. We added “a” before “zinc mining” in the sentence “[...] isolated from a metallicolous soil of zinc mining area in Languedoc, France reported by Vidal et al. [14]”.

Comment 27) Replace “our” with “Our”, add a “the” before “potential” and delete the tab between “phytoremediation” and “/” and “/” and “bioremediation” in the sentence “our strain might have potential for phytoremediation / bioremediation of HMs similar to some HMs resistant bacteria such as Mesorhizobium sp. loti HZ76 and Mesorhizobium cicero, both of which have demonstrated to improve phytoremediation [36,56]”.

-Thank you for the suggestion. We replaced “our” with “Our” and also added a “the” before “potential” and deleted the tab between “phytoremediation” and “/” and “/” and “bioremediation” in the sentence “our strain might have potential for phytoremediation / bioremediation of HMs similar to some HMs resistant bacteria such as Mesorhizobium sp. loti HZ76 and Mesorhizobium cicero, both of which have demonstrated to improve phytoremediation [36,56]”.

Comment 28) Replace “tolerance for salt” with “salt tolerance”.

-Thank you for the suggestion. We replaced “tolerance for salt” with “salt tolerance”.

Comment 29) Add the legend of supplementary tables with the legend of supplementary figures.

-Thank you for the suggestion. We added the legend of supplementary tables with the legend of supplementary figures.

Supplementary Table 1) The specific epithet of “Lotus Japonicus” should start with a lowercase in the column “Host”.

-Thank you for the suggestion. We changed the epithet of “Lotus Japonicus” with “Lotus japonicus”

Supplementary Tables 1 and 2) The superscript T of type strains should not be between parentheses in the columns “Organism” and “Strain name”, respectively.

-Thank you for the suggestion. We removed the parenthesis and wrote only the superscript T of type strains should not be between parentheses in the columns “Organism” and “Strain name”, respectively.
